# Structural characterization of human RPA70N association with DNA damage response proteins

Yeyao Wu[†], Wangmi Fu[†], Ning Zang[†], Chun Zhou*

School of Public Health & Sir Run Run Shaw Hospital, Zhejiang University School of Medicine, Zhejiang, China

**Abstract** The heterotrimeric Replication protein A (RPA) is the ubiquitous eukaryotic single-stranded DNA (ssDNA) binding protein and participates in nearly all aspects of DNA metabolism, especially DNA damage response. The N-terminal OB domain of the RPA70 subunit (RPA70N) is a major protein-protein interaction element for RPA and binds to more than 20 partner proteins. Previous crystallography studies of RPA70N with p53, DNA2 and PrimPol fragments revealed that RPA70N binds to amphipathic peptides that mimic ssDNA. NMR chemical-shift studies also provided valuable information on the interaction of RPA70N residues with target sequences. However, it is still unclear how RPA70N recognizes and distinguishes such a diverse group of target proteins. Here, we present high-resolution crystal structures of RPA70N in complex with peptides from eight DNA damage response proteins. The structures show that, in addition to the ssDNA mimicry mode of interaction, RPA70N employs multiple ways to bind its partners. Our results advance the mechanistic understanding of RPA70N-mediated recruitment of DNA damage response proteins.

**\*For correspondence:**
chunzhou@zju.edu.cn

[†]These authors contributed equally to this work

**Competing interest:** The authors declare that no competing interests exist.

## Editor's evaluation

This important paper advances our understanding of how a eukaryotic single-stranded DNA binding protein, Replication protein-A (RPA) interacts with multiple proteins in DNA transactions. The author provided compelling structure information on an OB-fold called RPA70N (or DBD-F) with 8 different peptides from various DNA metabolisms, which is complemented by in vivo studies. This paper will be of interest to researchers in DNA replication, recombination, and repair as well as structural biologists interested in a weak protein-protein interaction.

## Introduction

Replication protein A (RPA) is a heterotrimeric protein complex composed of the RPA70, RPA32 and RPA14 subunits (*Figure 1A*; *Fairman and Stillman, 1988*; *Wood et al., 1988*). It is the major eukaryotic single-stranded DNA (ssDNA) binding protein and has an affinity for ssDNA in the range of $10^{-9}$–$10^{-10}$ M (*Blackwell and Borowiec, 1994*; *Iftode et al., 1999*; *Kim et al., 1994*; *Kim et al., 1992*; *Wold, 1997*). Several DNA-binding domains (DBD-A, B, C, D), which are also called oligonucleotide binding domains (OB), form the core ssDNA binding region (*Figure 1A*; *Bochkarev et al., 1997*; *Bochkareva et al., 2002*; *Fan and Pavletich, 2012*; *Flynn and Zou, 2010*; *Murzin, 1993*; *Yates et al., 2018*). Due to its high affinity for ssDNA, RPA is involved in almost all aspects of DNA replication, repair, and recombination (*Caldwell and Spies, 2020*; *Chen and Wold, 2014*; *Fanning et al., 2006*; *Iftode et al., 1999*; *Maréchal and Zou, 2015*; *Wold, 1997*; *Zou et al., 2006*). It helps to protect ssDNA from nucleolytic degradation and prevents ssDNA entanglement by removing DNA secondary structures.

In addition to its ssDNA binding function, RPA also serves as a beacon to recruit a plethora of protein factors that are involved in DNA metabolism, mostly through the RPA70N and RPA32C (winged-helix) domains (*Figure 1A*; *Awate and Brosh, 2017*; *Caldwell and Spies, 2020*; *Fanning et al., 2006*; *Maréchal and Zou, 2015*). The RPA70N domain adopts an OB fold with a five-stranded anti-parallel beta-barrel but has very weak ssDNA affinity (*Figure 1B*; *Jacobs et al., 1999*). Its primary role is to mediate protein–protein interaction with its basic and hydrophobic groove and a side pocket (*Figure 1C*), as first shown by a series of studies of RPA70N interacting with p53 (*Abramova et al., 1997*; *Dutta et al., 1993*; *Li and Botchan, 1993*). The groove is flanked by two protruding loops, namely L12 and L45 (*Figure 1B and C*). The residues forming the groove and the side pocket are highly conserved, indicating that they are critical for RPA70N's protein interaction role (*Figure 1D and E*; *Yariv et al., 2023*). In the crystal structure of the RPA70N–p53 complex, the acid-hydrophobic peptide of p53 is shown to interact with the complementary basic and hydrophobic groove, mimicking ssDNA binding to OB domains (*Figure 1—figure supplement 1A*; *Bochkareva et al., 2005*).

RPA70N binds to the p53 transactivation domain to coordinate DNA repair with the p53-dependent checkpoint control (*Dutta et al., 1993*). RPA70N also binds to the N-terminus of ATRIP and is responsible for recruiting the ATR–ATRIP complex to DNA damage sites to initiate the cell-cycle checkpoint (*Ball et al., 2005*; *Namiki and Zou, 2006*; *Zou and Elledge, 2003*). Later it was shown that RPA70N mediates the interaction of RPA with the MRN complex and the 9-1-1 complex to protect replication forks during the DNA damage response, through binding to MRE11 and RAD9 (*Oakley et al., 2009*; *Olson et al., 2007*; *Robison et al., 2004*; *Wu et al., 2005*; *Xu et al., 2008b*). Recently, studies identified Ewing Tumor-associated Antigen 1 (ETAA1) as a DNA replication stress response protein and an ATR activator, which interacts with both RPA70N and RPA32C (*Bass et al., 2016*; *Feng et al., 2016*; *Haahr et al., 2016*; *Lee et al., 2016*). Besides cell cycle regulatory proteins, many helicases that are involved in DNA repair interact with RPA70N (*Awate and Brosh, 2017*). Both BLM (Sgs1) and DNA2 interact with RPA and form a complex to carry out long-range DNA resection during double-strand DNA break repair (*Cejka et al., 2010*; *Gravel et al., 2008*; *Nimonkar et al., 2011*; *Zhu et al., 2008*). RPA was proposed to recruit both DNA2 and BLM through RPA70N, stimulating the helicase activity of BLM while enhancing the nuclease activity of DNA2 by removing DNA secondary structures (*Brosh et al., 2000*; *Doherty et al., 2005*; *Nimonkar et al., 2011*; *Zhou et al., 2015*). WRN, HelB and FancJ also bind to RPA through RPA70N, and the presence of RPA greatly enhanced their helicase activities (*Brosh et al., 1999*; *Doherty et al., 2005*; *Guler et al., 2012*; *Gupta et al., 2007*; *Hormeno et al., 2022*; *Shen et al., 1998*; *Shen et al., 2003*; *Suhasini et al., 2009*; *Tkáč et al., 2016*). Moreover, many other proteins that are involved in DNA repair and replication also interact with RPA70N. For example, the RMI1 component of the BTR complex (BLM–Topo IIIα–RMI1–RMI2) and PrimPol (DNA primase and DNA polymerase) directly associate with RPA70N (*Dornreiter et al., 1992*; *Guilliam et al., 2017*; *Shorrocks et al., 2021*; *Wan et al., 2013*; *Xue et al., 2013*).

In general, most of the proteins that interact with RPA70N utilize a motif of around 20 amino acids long that contains a mixture of acidic and hydrophobic residues (*Shorrocks et al., 2021*). The exact sequence of these motifs doesn't share much homology, despite the similarity in the overall composition, indicating that each motif might bind to RPA70N differently (*Figure 1—figure supplement 1A*). To better understand the mechanism of RPA70N-mediated target protein recruitment, we set out to determine the complex structures of RPA70N with the peptide motifs that bind to it. To date, quite a few studies have employed NMR chemical shifts to probe the interaction sites of RPA70N with partner proteins (*Guler et al., 2012*; *Kang et al., 2018*; *Liu et al., 2011*; *Ning et al., 2015*; *Xu et al., 2008b*; *Yeom et al., 2019*). The NMR chemical shift information is useful in identifying potential residues that are involved in binding, but owing to the transient nature of the interactions, the complex structures were not resolved by NMR. Several crystal structures of RPA70N in complex with bound peptide have been reported, namely those of RPA70N–p53, RPA70N–DNA2, RPA70N–PrimePol and Rfa1N–Ddc2 (*Bochkareva et al., 2005*; *Deshpande et al., 2017*; *Guilliam et al., 2017*; *Zhou et al., 2015*). However, crystallization attempts often yield crystals of RPA70N itself without peptide bound, mainly owing to the weak affinity between RPA70N and the protein sequences that it recognizes (*Souza-Fagundes et al., 2012*). To overcome this problem, we fused each target sequence to the C-terminus of RPA70N with a flexible linker. By adjusting the sequence and the linker length of the interacting peptides, we managed to crystalize and determine the structures of RPA70N in complex with HelB,

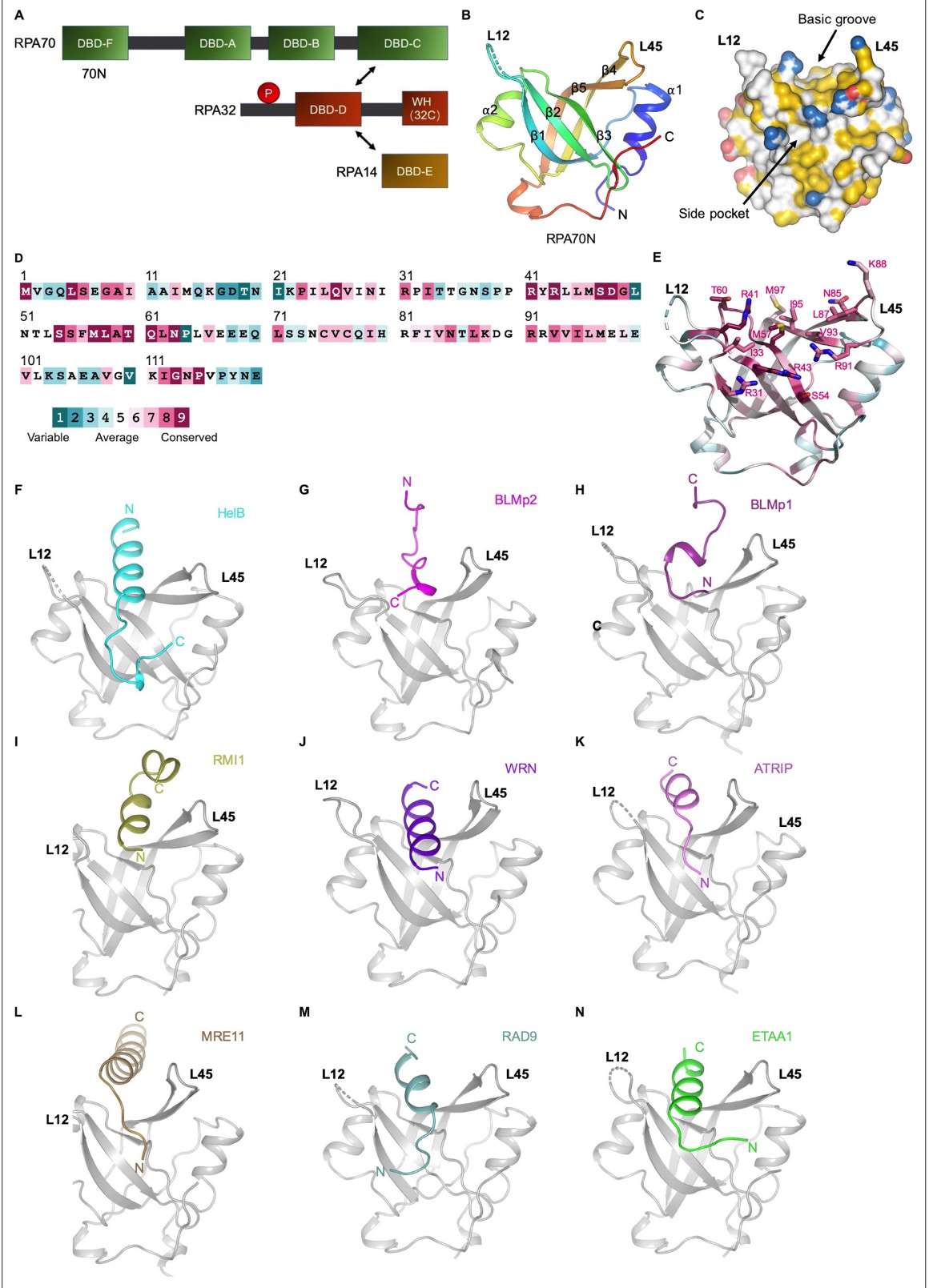

**Figure 1.** RPA70N–peptide complex structures determined in this study. (**A**).Linear domain diagram of the RPA heterotrimer. (**B**) Ribbon representation of human RPA70N from PDB 5EAY. L12 denotes the loop between β1 and β2, L45 is the loop between β4 and β5. (**C**) Surface representation of RPA70N from PDB 5EAY, showing the basic groove and the side pocket, with hydrophobic, positively and negatively charged atoms in yellow, blue and red, respectively, and other atoms in white. (**D**) Conservation of the human RPA70N sequence. (**E**) Most of the conserved residues in RPA70N are located at

*Figure 1 continued on next page*

**eLife** Research article

Biochemistry and Chemical Biology | Structural Biology and Molecular Biophysics

*Figure 1 continued*

the basic groove region. (**F–N**) Ribbon representation of the RPA70N–peptide complex structures determined in this study displayed in the same view. RPA70N is colored in gray.

The online version of this article includes the following source data and figure supplement(s) for figure 1:

**Source data 1.** Data collection and refinement statistics.

**Figure supplement 1.** Sequence alignment and omit maps of bound peptides (related to *Figure 1*).

BLM, RMI1, WRN, ATRIP, MRE11, RAD9, and ETAA1 (*Figure 1F–N*, *Figure 1—figure supplement 1B–K*, *Figure 1—source data 1*).

## Results

### Structure of the RPA70N–HelB peptide complex

HelB is a conserved helicase that is involved in DNA replication initiation, in replication stress responses and in negatively regulating DNA end-resection (*Guler et al., 2012*; *Hazeslip et al., 2020*; *Taneja et al., 2002*; *Tkáč et al., 2016*). It has an RPA-binding motif located in the helicase domain and its recruitment to chromatin correlates with the level of RPA (*Guler et al., 2012*). A recent in vitro study showed that all ssDNA-dependent activities of HelB are greatly stimulated by RPA–ssDNA filaments (*Hormeno et al., 2022*). We crystallized a human HelB helicase peptide (HelBp1, residues 496–519) with the human RPA70N (residues 1–120) using the fusion strategy in space group P41212 and found that there is one molecule in the asymmetry unit (*Figures 1F and 2B*, *Figure 1—source data 1*). Crystal packing analysis shows that the HelB peptide from one fusion protein is bound by a neighboring RPA70N molecule (*Figure 2—figure supplement 1A*). The electron density of the fusion linker is not observed as it is highly flexible. In the 1.6 Å structure, residues 496–517 of HelB form a four-turn α helix followed by a β turn and a $3_{10}$ helix (*Figures 1F and 2B*). The curved β sheet of RPA70 and the extending L12, L45 loops form a shallow groove where the amphipathic helix of HelB sits (*Figures 1F and 2B*). The negatively charged residues E496, E499 and D506 of HelB form hydrogen bonds and salt bridges with RPA70N R81, T60, Q61 and R41 (*Figure 2B and C*). The hydrophobic residues V500, C504 and F507 of HelB pack against a broad hydrophobic patch formed by RPA70N I83, M97, I95, V93, L87 and M57 (*Figure 2C*). The mixed basic and hydrophobic character of the RPA70N groove complements the acidic-hydrophobic nature of the HelB peptide (*Figure 2B and C*). The interacting residues correlate well with the results of the previous NMR chemical shift analysis and mutation studies regarding the charged and hydrophobic residues (*Guler et al., 2012*). On the left side of the groove, W517 of HelB fits into a well-defined pocket (side-pocket) formed by the aliphatic portions of RPA70N N29, R31, R43 and S54 (*Figures 1C, 2D and E*). HelB residues D510, E516 and T519 were stabilized by hydrogen bonding or electrostatic interactions with the side chains of RPA70N R31, N29 and R91. In addition, RPA70N R43 forms a hydrogen bond with the main chain carbonyl group of HelB W517, further stabilizing the folded-back conformation of the HelB peptide (*Figure 2E*). The overall binding mode of HelB to RPA70N is similar to that of DNA2 (*Figure 2F*). The amphipathic helix of HelB overlaps with the DNA2 helix while the β turn coincides with the β turn region of DNA2. Both peptides have a conserved hydrophobic residue that fits into the side pocket (*Figure 2F*). ITC titration results showed that mutation of W517 in HelB to alanine reduced the affinity between HelB peptide and RPA70N from around 4 μM to 16 μM (*Figure 2G* and *Figure 2—figure supplement 1B–1D*), highlighting the contribution of side-pocket interactions to the overall binding strength. Further mutation of the hydrophobic residues F507 and C504 in HelB reduced the affinity between RPA70N and HelB by ten fold (37 μM). In HeLa cells, these mutations led to reduced colocalization of HelB–EGFP and endogenous RPA in untreated cells ls and in camptothecin (CPT) or hydroxyurea (HU) treated cells to different degrees (*Figure 2H–J*, *Figure 2—figure supplement 1E and F*). HelB-EGFP with C504A/F507A/W517A triple mutation almost completely lost the ability to form obvious RPA foci (*Figure 2H–J*, *Figure 2—figure supplement 1E and F*). Coimmunoprecipitation (co-IP) analysis also showed that triple mutant HelB-EGFP pulled down much less RPA than WT (*Figure 2K*). In addition, ITC titration showed that mutation of the conserved arginine esidues in RPA70N (R31, R41, R43, R91) significantly reduced the binding of RPA70N to

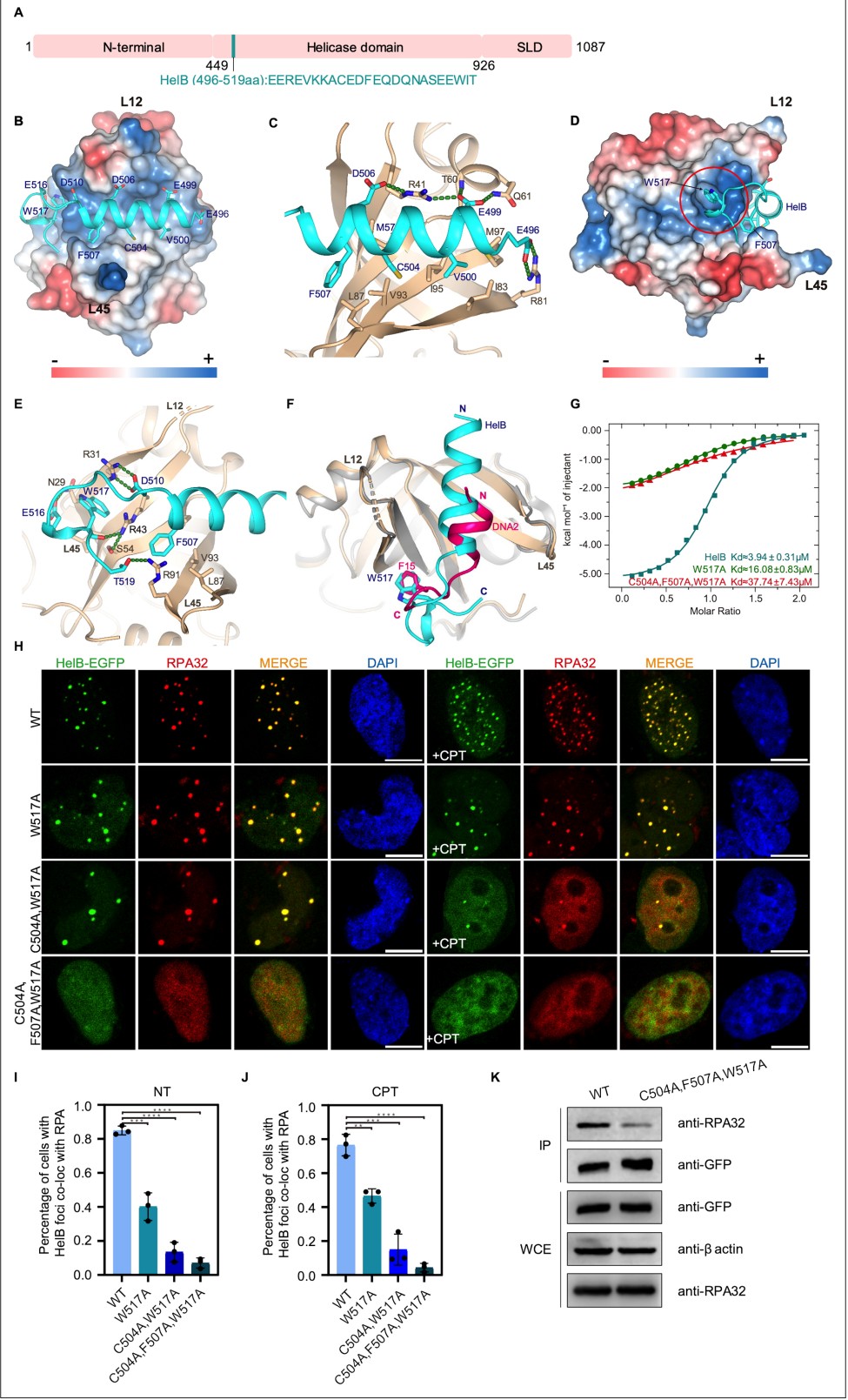

**Figure 2.** Structure of the RPA70N–HelB complex. (**A**) Linear domain diagram of HelB showing the position and sequence of the RPA70N interacting motif. SLD: subcellular domain. (**B**) Structure of RPA70N in complex with HelB, showing the surface charges of RPA70N; hydrophobic and negatively charged residues of HelB are displayed as sticks, the HelB peptide is colored in cyan. (**C**) Close-up view of the amphipathic HelB helix interacting with the

*Figure 2 continued on next page*

*Figure 2 continued*

groove of RPA70N. HelB peptide is colored in cyan and RPA70N in beige. Green dashed lines indicate hydrogen bonds or salt bridges. (**D**) Side view of the RPA70N–HelB structure, the electrostatic surface of RPA70N is displayed and the side pocket is highlighted. The representation is rotated 90° compared to (**B**). (**E**) Close-up view of the side-pocket residues coordinating the C-terminal part of the HelB peptide. Green dashed lines indicate hydrogen bonds or salt bridges. (**F**) Superposition of RPA70N–HelB structure with the RPA70N–DNA2 structure (PDB:5EAY). HelB is colored in cyan, DNA2 in light-magenta. The direction of the HelB and DNA2 peptides in the RPA70N groove is the same in both structures. Both proteins have a hydrophobic residue inserted into the side pocket of RPA70N. (**G**) Isothermal titration calorimetry (ITC) results for WT HelB (496–519aa), W517A or C504A/F507A/W517A mutant peptides with RPA70N. (**H**) HeLa cells expressing HelB–EGFP, HelB (W517A)–EGFP, HelB (C504A/W517A)–EGFP, or HelB (C504A/F507A/W517A)–EGFP were treated with medium control (NT) or camptothecin (CPT) (2 μM, 2 h), fixed and immunostained with an anti-RPA32 antibody. The scale bar is 10 μm. (**I**) and (**J**). Quantification of data from (**H**). Data are presented as mean ±s.d. of three independent experiments. 100 cells from each experiment were analyzed, and cells containing more than three bright HelB and RPA co-localization foci were defined as positive. Statistical analysis was performed using a two-tailed Student's t-test (**** $P<0.0001$, *** $P<0.001$, ** $P<0.01$, * $P<0.05$). (**K**) Immunoprecipitation and western blot showing that mutation of HelB residues reduced RPA association. Anti-EGFP magnetic beads were used to carry out immunoprecipitations, followed by probing with an anti-RPA32 antibody. IP, immunoprecipitation; WCE, whole cell extract.

The online version of this article includes the following source data and figure supplement(s) for figure 2:

**Source data 1.** Raw data of all western blots from *Figure 2*.

**Figure supplement 1.** Characterization of RPA70N–HelB interaction (related to *Figure 2*).

---

HelBp1 (*Figure 2—figure supplement 1G–J*). Together, these results revealed the molecular basis for the HelB-RPA70N interaction.

## Structures of the RPA70N–BLM peptide complexes

BLM helicase is a multifunctional RecQ family helicase. It is involved in DNA-end resection, restart of stalled replication forks, dissolving Holliday junctions, and processing of ultra-fine DNA bridges (*Bythell-Douglas and Deans, 2021*; *Chu and Hickson, 2009*; *Croteau et al., 2014*; *Kitano, 2014*; *Shorrocks et al., 2021*). It has two RPA70N binding motifs in the N-terminal disordered region, namely residues 146–165 (BLMp1) and residues 550–570 (BLMp2) (*Figure 3A*; *Doherty et al., 2005*; *Shorrocks et al., 2021*). Shorrocks and coworkers reported that RPA and BLM proteins accumulated along laser lines within 5 min but at relatively low levels, RPA microfoci appeared in ~50% of irradiated cells after 15 min as a result of DNA end-resection in S and G2 cells. Shortly afterwards, at 20 min, BLM microfoci appeared and co-localized with RPA; whereas a BLM mutant lacking both BLMp1 and BLMp2 failed to form microfoci (*Shorrocks et al., 2021*). We fused BLMp1 and BLMp2 separately to RPA70N and determined the structures of these regions (*Figure 1G and H*).

In the structure of RPA70N–BLMp2, BLM residues 550–564 are visible and the peptide is bound by two RPA70N molecules (*Figure 3A and B*, *Figure 3—figure supplement 1A*). The C-terminal part of the kinked BLM peptide fits onto the RPA70N groove, with F556, I558 and F561 making contacts with the hydrophobic patch of RPA70N, while RPA70N residues R41, K88, R91 and R43 form salt bridges or hydrogen bonds with BLM D560, D552, D562 and the main chain carbonyl group of F561 (*Figure 3C*). Interestingly, the N-terminal half of BLMp2 latches onto the α1 region of a nearby RPA70N[b] (*Figure 3D*). K16[b] from RPA70N[b] forms several ionic interactions with BLM D554, D557 and D559 to neutralize the negative charges. Q15[b] also contributes to the interaction by forming two hydrogen bonds with BLM D557. Near the tip of the BLMp2 peptide, Y551 fits onto a small hydrophobic surface formed by A9[b], A12[b], I13[b] and I21[b] in RPA70N[b], its main chain amide group also forms a hydrogen bond with the carbonyl group of E7[b] (*Figure 3D*). Overall, it appears that BLMp2 promotes Loop 12 of one RPA70N dimer to interact with α1[b] of the other RPA70N, and that each RPA70N provides some of the binding surface for BLMp2 (*Figure 3D* and *Figure 3—figure supplement 1A*). Using ITC, we found that BLMp2 binds to RPA70N with a relatively weak $K_D$ of around 18 μM, and that mutation of BLMp2 residues (D560A, F561A, D562A) almost abolished the binding between BLMp2 and RPA70N (*Figure 3E*, *Figure 3—figure supplement 1B-D*).

In the structure of RPA70N–BLMp1, BLMp1 also adopts a kinked conformation and bridges two RPA70N molecules (*Figure 3F*). However, one major difference is that BLMp1 binds to RPA70N in the reverse direction when compared to BLMp2, HelB or DNA2 (*Figure 1F–H*). The N-terminal part of

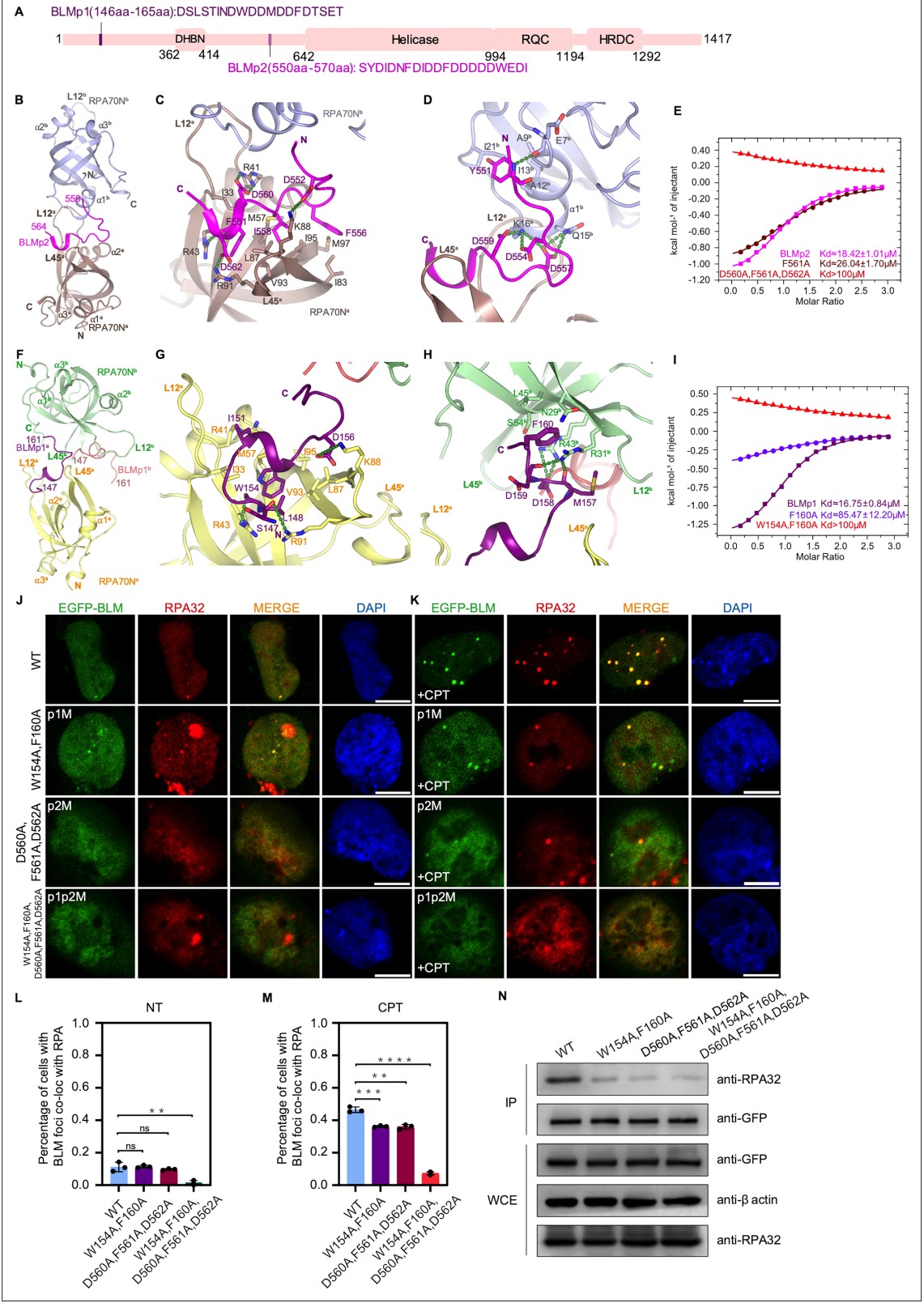

**Figure 3.** Structures of two RPA70N–BLM complexes. (**A**) Linear domain diagram of BLM showing the position and sequence of RPA70N interacting motifs. DHBN, dimerization helical bundle in N-terminal region; RDC, RecQ-conserved domain; HRDC, helicase and RNaseD in C-terminal region. (**B**) Ribbon representation of the RPA70N–BLMp2 crystal structure, the BLMp2 peptide is coordinated by two RPA70N molecules. BLMp2 is colored in magenta and the two RPA70N molecules are colored in brown and light-blue, respectively. (**C**) Close-up view of BLMp2 interacting with the RPA70Na

*Figure 3 continued on next page*

*Figure 3 continued*

groove. Interacting residues are shown as sticks, green dashed lines indicate hydrogen bonds or salt bridges. (**D**) Close-up view of BLMp2 interacting with RPA70Nb α1, interacting residues are shown as sticks, green dashes indicate hydrogen bonds or salt bridges. (**E**) ITC titration results for WT BLMp2 (550–570aa), F561A or D560A/F561A/D562A mutant peptide with RPA70N. (**F**) Ribbon representation of the RPA70N–BLMp1 crystal structure: the BLMp1 peptide is coordinated by two RPA70N molecules. BLMp1[a] is colored in purple, BLMp1[b] is colored in pink and the two RPA70N molecules are colored in yellow and light-green, respectively. (**G**) Close-up view of BLMp1 interacting with the RPA70Na groove. Interacting residues are shown as sticks, green dashes indicate hydrogen bonds or salt bridges. (**H**) Close-up view of BLMp1 interacting with the RPA70Nb side-pocket. Interacting residues are shown as sticks, green dashes indicate hydrogen bonds or salt bridges. (**I**) ITC titration results for WT BLMp1 (146–165aa), F160A or W154A/F160A mutant peptide with RPA70N. (**J, K**) HeLa cells expressing EGFP–BLM, EGFP–BLM(W154A/F160A, p1M), EGFP–BLM (D560A/F561A/D562A, p2M), or EGFP–BLM(W154A/F160A/D560A/F561A/D562A, p1p2M) were treated with medium control (NT) or CPT (2 μM, 2 h), fixed and immunostained with an anti-RPA32 antibody. The scale bar is 10 μm. (**L, M**). Quantification of data from (**J**) and (**K**). Data are presented as mean ± s.d. of three independent experiments. 100 cells from each experiment were analyzed, cells containing more than three bright BLM and RPA co-localization foci were defined as positive. Statistical analysis was performed using a two-tailed Student's t-test (**** $P<0.0001$, *** $P<0.001$, ** $P<0.01$, * $P<0.05$). (**N**) Immunoprecipitation and western blot showing that mutation of BLM residues reduced RPA association. Anti-EGFP magnetic beads were used to carry out immunoprecipitations, followed by probing with an anti-RPA32 antibody. IP, immunoprecipitation; WCE, whole cell extract.

The online version of this article includes the following source data and figure supplement(s) for figure 3:

**Source data 1.** Raw data from all western blots shown in *Figure 3*.

**Figure supplement 1.** Characterization of RPA70N–BLM interaction (related to *Figure 3*).

**Figure supplement 2.** Characterization of the RPA70N–BLM interaction (related to *Figure 3*).

BLMp1 forms a one-turn helix followed by a γ turn (*Figure 3G*). BLM W154 inserts into a hydrophobic pocket formed by RPA70N V93, I95, M57, I33 and the aliphatic part of R43, and stacks with M57. BLM L148 stacks on top of W154, and BLM I151 packs onto the side chains of RPA70N I33, M57 and R41 (*Figure 3G*). At the middle of the BLMp1 peptide, D156 interacts with RPA70N K88 and the peptide forms another β turn. The C-terminal part of BLMp1 adopts an extended conformation, with F160 anchored in the side pocket of a nearby RPA70N[b] (*Figure 3H*). The RPA70N[b] residues R43[b], R31[b] also interact with the BLM D158 side chain and with main chain oxygen atoms of D159 and M157. The $K_D$ of the BLMp1–RPA70N complex, as determined by ITC, is around 16.7 μM, similar to that of the corresponding BLMp2 complex (*Figure 3I* and *Figure 3—figure supplement 1E*). Mutation of BLM F160 to alanine greatly reduced the affinity between BLMp1 and RPA70N, resulting in a $K_D$ of around 85 μM. F160A and W154A double mutation reduced the affinity even more (*Figure 3I*, *Figure 3—figure supplement 1F and G*). The two RPA70N molecules are connected by the BLMp1 peptide but do not make other contacts (*Figure 3F* and *Figure 3—figure supplement 1H*). The way that BLMp1 bridges two RPA70N molecules is analogous to the role of the p53 peptide (*Figure 3—figure supplement 1I and J*). However, the direction of the BLMp1 peptide in the groove is reversed compared to that of the p53 peptide (*Figure 3—figure supplement 1J*).

In untreated HeLa cells, WT or mutant EGFP–BLM formed few foci (*Figure 3J*). CPT treatment increased the percentage of cells with obvious foci to around 50% for WT and 40% for BLMp1 or BLMp2 mutants (*Figure 3K*). However, the BLMp1 or BLMp2 mutants generally displayed fewer foci per cell when compared to WT (*Figure 3J–M*). The BLMp1 and BLMp2 dual mutant formed far fewer foci, even after CPT treatment (*Figure 3J–M*). Similar phenomena were observed for HU-treated cells (*Figure 3—figure supplement 2A and B*). Co-IP experiments showed that mutating the RPA70N interacting residues in BLM reduced its binding to RPA (*Figure 3N*). These results are in agreement with the results of Shorrocks, who reported that BLM has two RPA70N interacting motifs (*Shorrocks et al., 2021*). Not surprisingly, mutation of the conserved arginine residues in RPA70N (R31, R41, R43, R91) significantly reduced the binding of RPA70N to BLMp2 and BLMp1 (*Figure 3—figure supplement 2C–I*).

## Structure of the RPA70N–RMI1 peptide complex

RMI1 is another RPA partner. It is a subunit in the BTR complex and mainly mediates protein–protein interaction (*Shorrocks et al., 2021*; *Wang et al., 2010*; *Xu et al., 2008a*; *Xue et al., 2013*). The RPA70N interaction motif is located between its two OB folds (*Figure 4A*). In the RPA70N–RMI1 complex structure, RMI1 residues 243-259 form two short α helixes with a β turn in the middle (*Figures 4B and 1I*, *Figure 4—figure supplement 1A*). The overall arrangement is similar to the complex of RPA70N–BLMp1 with the N-terminal helix in the groove and the C-terminal helix binding

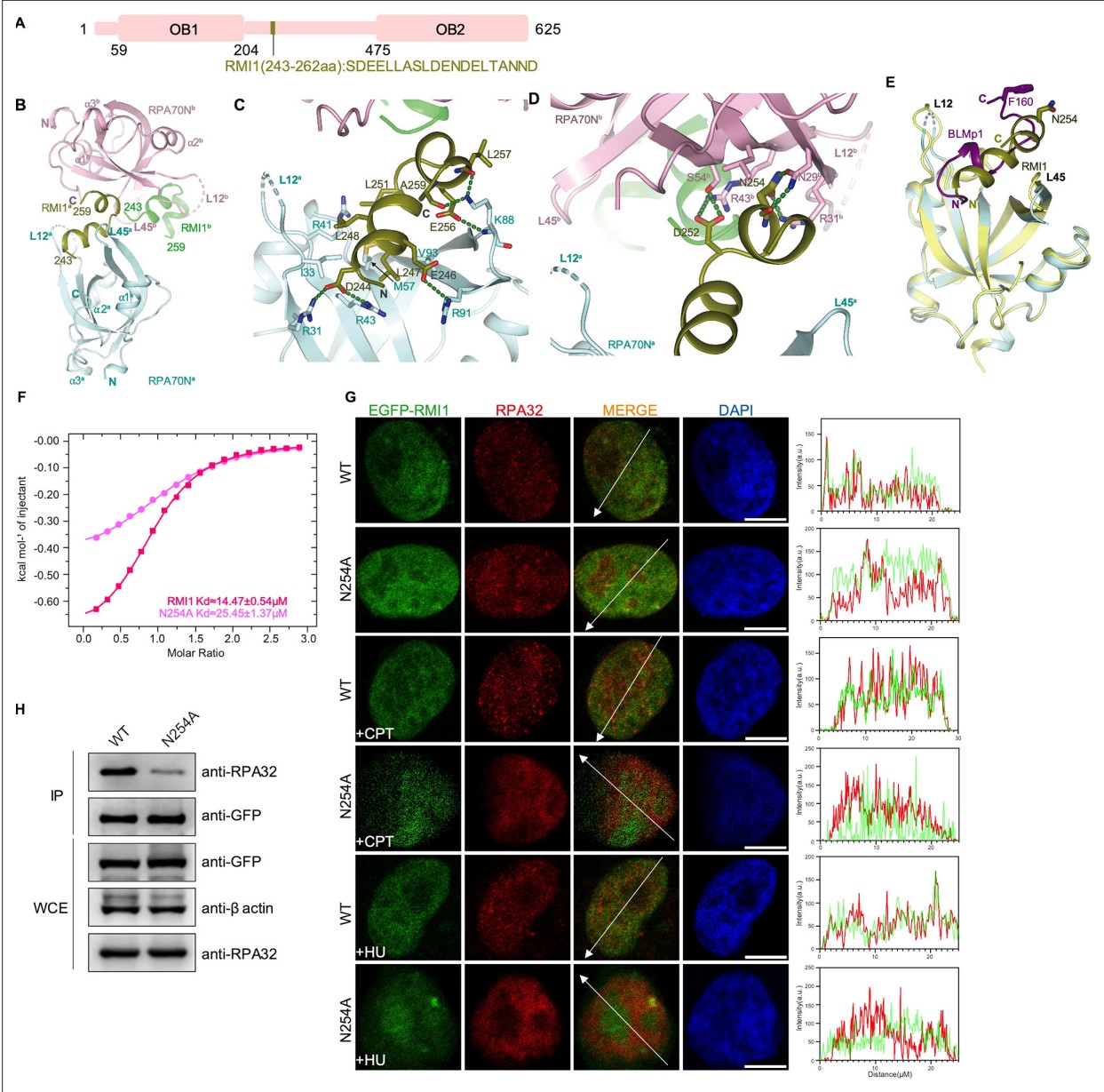

**Figure 4.** Structure of the RPA70N–RMI1 complex. (**A**) Linear domain diagram of RMI1, showing the position and sequence of the RPA70N interacting motif. (**B**) Ribbon representation of the RPA70N–RMI1 crystal structure, the RMI1 peptide is coordinated by two RPA70N molecules. RMI1 is colored in olive and the two RPA70N molecules are colored pale-cyan and light-pink. The other RMI1 peptide in RPA70Nb is colored green. (**C**) Close-up view of RMI1 interacting with the RPA70Na groove, interacting residues are shown as sticks, green dashed lines indicate hydrogen bonds or salt bridges. (**D**) Close-up view of RMI1 interacting with the RPA70Nb side-pocket. Interacting residues are shown as sticks, and green dashed lines indicate hydrogen bonds or salt bridges. (**E**) Superposition of the RPA70N–RMI1 structure with that of the RPA70N–BLMp1 complex: RMI1 and BLMp1 interact with RPA70N in a similar manner. (**F**) ITC titration results for the WT RMI1 peptide (243–262aa) or the N254A mutant peptide with RPA70N. (**G**) HeLa cells expressing EGFP–RMI1 or EGFP–RMI1(N254A) were treated with medium control, CPT (2 μM, 2 h) or HU (2 mM, 3 h), fixed and immunostained with an anti-RPA32 antibody. The scale bar is 10 μm. The intensities of the fluorescent signals for EGFP–RMI1 or EGFP RMI1(N254A) and RPA32 are displayed on the right. (**H**) Immunoprecipitation and western blotting showed that mutation of RMI1 residues reduced the RPA association. Anti-EGFP magnetic beads were used to carry out immunoprecipitation, which was followed by probing with an anti-RPA32 antibody. IP, immunoprecipitation; WCE, whole cell extract.

The online version of this article includes the following source data and figure supplement(s) for figure 4:

**Source data 1.** Raw data for all all western blots shown in *Figure 4*.

**Figure supplement 1.** Characterization of the RPA70N–RMI1 interaction (related to *Figure 4*).

to a neighboring RPA70N[b] (*Figures 4B and 3F*). RMI1 L247 of the N-terminal helix fits into a hydrophobic pocket at the bottom of the RPA70N groove (*Figure 4C*). RMI1 L248 and L251 interact with the hydrophobic side chains of RPA70N I33, M57 and the aliphatic part of R41. RMI1 D244 and E246 are stabilized by electrostatic interactions with RPA70N R31, R43 and R91. The folded-back C-terminal helix also interacts with RPA70N K88 by forming several hydrogen bonds (*Figure 4C*). RMI1 N254 inserts into the side pocket of a neighboring RPA70N[b] and forms a few hydrogen bonds with N29[b] and R31[b] side chains (*Figure 4D*). D252 also interacts with S54[b] and R43[b], further strengthening the interaction (*Figure 4D*). Analogous to RPA70N–BLMp1, the two RPA70N molecules coordinating the RMI1 peptide aren't making any contact (*Figure 4—figure supplement 1A*). Superposition of the two structures showed that BLMp1 F160 and RMI1 N254 point in the same direction but not at the exact same location, indicating that the second RPA70N molecule is able to adjust to different peptide sequences for binding (*Figure 4E*). ITC titration showed that the RMI1 peptide binds to RPA70N with a $K_D$ of around 14.5 μM, mutation of N254 reduced the affinity between RMI1 and RPA70N to around 25.6 μM (*Figure 4F*, *Figure 4—figure supplement 1B and C*). In HeLa cells, the expression of WT or N254A EGFP–RMI1 resulted in very few foci. Nevertheless, N254A mutation led to less RPA-RMI1 colocalization as measured by fluorescent intensity analysis across the nucleus (*Figure 4G*). Accordingly, co-IP results showed reduced RPA interaction for the RMI1 N254A mutant (*Figure 4H*). Mutation of the conserved arginine residues in RPA70N (R31, R41, R43, R91) significantly reduced the binding of RPA70N to RMI1 (*Figure 4—figure supplement 1D–H*).

## Structure of the RPA70N–WRN peptide complex

WRN nuclease-helicase belongs to the RecQ family of DNA helicases and plays important roles in DNA repair and in the maintenance of genome integrity (*Chu and Hickson, 2009*; *Croteau et al., 2014*; *Kitano, 2014*; *Mukherjee et al., 2018*). Studies carried out by *Doherty et al., 2005* and *Lee et al., 2018* showed that RPA stimulates WRN helicase activity in a concentration-dependent manner and that the helicase activity of WRN requires the binding of multiple RPAs (*Lee et al., 2018*). WRN has two tandem RPA-binding motifs with the same sequence localized between its nuclease and helicase domains (*Doherty et al., 2005*; *Shen et al., 2003*; *Yeom et al., 2019*; *Figure 5A*). We fused one WRN motif to RPA70N, and the fusion construct crystallized in space group P212121 with two molecules in the asymmetry unit (*Figure 5B*, *Figure 1—source data 1*). In the structure, WRN 435–451 form a continuous α helix and insert into the amphipathic groove of a symmetry-related RPA70N[b] (*Figure 5B*). Residues E439-R31[b]-D443-R43[b]-E442-R91[b]-E445 form a series of electrostatic interactions, and WRN M446 and L449 contact the hydrophobic patch formed by RPA70N[b] L87[b], V97[b], I33[b], M57[b] and I95[b] (*Figure 5C*). The N-terminal part of each WRN peptide helix interacts with the RPA70N peptide that it fused to. WRN[b] Y436[b] fits into the side-pocket and forms two hydrogen bonds with RPA70N[b] R31[b] (*Figure 5D and E*). Compared to the RPA70N–p53 structure, the direction of the WRN[a] peptide in the groove is reversed and the positions of WRN Y436 and p53 M44 are different (*Figure 5—figure supplement 1A and B*). The direction of the other WRN[b] peptide bound to the side pocket is the same as that of p53[b] (*Figure 5—figure supplement 1A and B*). ITC titration of the WRN peptide with RPA70N yielded a $K_D$ value around 11.6 μM, whereas M466A mutation increased the $K_D$ value to around 37.4 μM (*Figure 5F and G*, *Figure 5—figure supplement 1C and D*).

In HeLa cells, M466A and M473A double mutants had reduced WRN–RPA colocalization, as well as a reduced percentage of cells with obvious RPA foci (*Figure 5H–K*). EGFP–WRN with M466A and M473A double mutation also pulled down less RPA. Mutation of the conserved arginine residues in RPA70N (R31, R41, R43, R91) greatly reduced the binding of RPA70N to the WRN peptide (*Figure 5—figure supplement 1E–1I*). These data jointly demonstrate that RPA70N-mediated interaction with WRN is critical for WRN recruitment during the DNA damage response.

## Structure of the RPA70N–ATRIP peptide complex

ATR is a member of the PIKK kinase family, and the ATR–ATRIP complex is a key regulator of the DNA damage checkpoint. The complex is recruited to DNA damage sites by RPA coated ssDNA through ATRIP (*Ball et al., 2005*; *Zou and Elledge, 2003*). We crystallized the RPA70N–ATRIP fusion protein in the P212121 space group with one molecule in the asymmetric unit (*Figure 1K*, *Figure 6A and B*, *Figure 1—source data 1*). ATRIP peptide binds to the RPA70N it fused to and the linker region is disordered. In the structure, ATRIP residues 53–69 form a three-turn helix with two short flanking loops

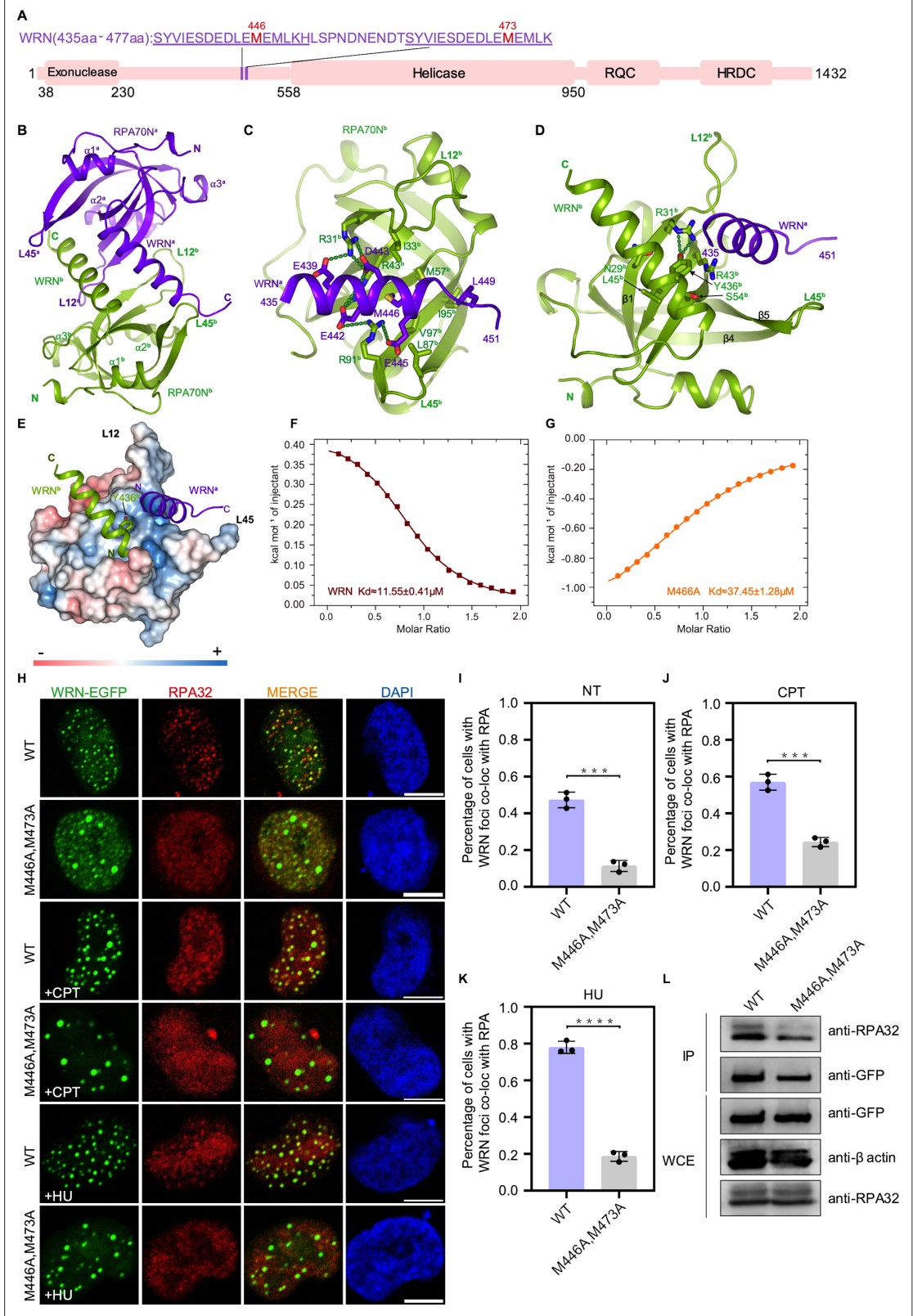

**Figure 5.** Structure of the RPA70N–WRN complex. (**A**) Linear domain diagram of WRN showing the position and sequence of RPA70N-interacting motifs. (**B**) Ribbon representation of the RPA70N–WRN crystal structure. The fused WRN peptide forms an α helix and inserts into the groove of the symmetry-related RPA70N molecule. The two RPA70N molecules and linked WRN peptides are colored in purple-blue and light-green, respectively. (**C**) Close-up view of WRN[a] interacting with the RPA70N[b] groove, interacting residues are shown as sticks, green dashed lines indicate hydrogen bonds

*Figure 5 continued on next page*

*Figure 5 continued*

or salt bridges. (**D**) Close-up view of WRN[b] interacting with the RPA70N[b] side-pocket. Interacting residues are shown as sticks, whereas green dashed lines indicate hydrogen bonds or salt bridges. (**E**) Y436[b] from WRN[b] inserts into the side pocket of RPA70N[b] while WRN[a] sits in the basic groove of RPA70N[b]. (**F**) ITC titration data for the WT WRN peptide (435–451aa) with RPA70N: the titration appears to be endothermic. (**G**) ITC titration data for the M466A mutant WRN peptide (435–451aa) with RPA70N. (**H**) HeLa cells expressing WRN–EGFP or WRN (M446A/M473A)–EGFP were treated with medium control (NT), CPT (2 μM, 2 h) or HU (2 mM, 3 h), fixed and immunostained with an anti-RPA32 antibody. The scale bar is 10 μm. (**I, J and K**) Quantification of data from (**H**), data are presented as mean ± s.d. of three independent experiments. 100 cells from each experiment were analyzed. Cells containing more than three bright WRN and RPA co-localization foci were defined as positive. Statistical analysis was performed using a two-tailed Student's t-test (**** $P<0.0001$, *** $P<0.001$, ** $P<0.01$, * $P<0.05$). (**L**) Immunoprecipitation and western blot showing that the mutation of WRN residues reduced RPA association. Anti-EGFP magnetic beads were used to carry out immunoprecipitation, which was followed by probing with an anti-RPA32 antibody. IP, immunoprecipitation; WCE, whole cell extract.

The online version of this article includes the following source data and figure supplement(s) for figure 5:

**Source data 1.** Raw data for all of the western blots shown in *Figure 5*.

**Figure supplement 1.** Characterization of the RPA70N–WRN interaction (related to *Figure 5*).

(*Figure 6A and B*). The hydrophobic side of the ATRIP helix, consisting of F55, L60, L63 and L66, packs against the broad hydrophobic patch of the RPA70N groove (*Figure 6B*). RPA70N R43 and R91 form salt-bridges and hydrogen bonds with ATRIP D54 and main-chain carbonyl groups at the N-terminus of the ATRIP peptide. At the C-terminus of the peptide, RPA70N R41 forms a hydrogen bond with the carbonyl group of ATRIP L63 while ATRIP E62 forms a hydrogen bond with the main-chain amide group of RPA70N K88. The direction of the ATRIP peptide is inverted compared to HelB or DNA2, instead it is the same as that seen in the *Kluyveromyces lactis* Ddc2 (ATRIP)–Rfa1N complex (PDB: 5OMB) (*Deshpande et al., 2017*), with both Ddc2 and ATRIP using a hydrophobic residue (F55 in ATRIP or I14 in Ddc2) at the N-terminus to anchor the peptide at the groove (*Figures 6C, 1F and K*). Aiming to inhibit the ATRIP–RPA70N interaction in cells and based on the structure of the RPA70N–p53 complex, *Frank et al., 2014* engineered a stapled helix peptide that binds to RPA70N and determined the co-crystal structure of the synthetic helix with RPA70N. In their structure (PDB:4NB3), the peptide is in a reversed orientation when compared to our structure or the Ddc2–Rfa1N structure and employs a 3,4-dichloro-substituted phenylalanine (ZCL) to bind the hydrophobic pocket that is bound by F55 in ATRIP (*Figure 6D*). Mutation of F55 to alanine greatly reduced the affinity of ATRIP towards RPA70N, indicating that the hydrophobic interactions mediated by F55 are critical for maintaining ATRIP–RPA70N association (*Figure 6E*, *Figure 6—figure supplement 1A and B*). F55A/L60A/L63A triple mutation almost abolished binding between the ATRIP peptide and RPA70N (*Figure 6E*, *Figure 6—figure supplement 1C*). In HeLa cells, WT ATRIP and RPA colocalized very well (*Figure 6F*). The ATRIP F55A single mutant displayed slightly weakened colocalization, whereas the F55A/L60A/L63A triple mutant showed worse colocalization (*Figure 6F–J*). Similarly, F55A and F55A/L60A/L63A pulled down less RPA than WT in co-IP experiments (*Figure 6K*). Mutation of the conserved arginine residues in RPA70N (R41, R43, R91) significantly reduced the binding of RPA70N to ATRIP (*Figure 6—figure supplement 1D–1F*). Our results provided the structural basis for RPA70N-mediated ATRIP/ATR recruitment, which is a crucial response to DNA damage.

## Structure of the RPA70N–MRE11 peptide complex

The MRN complex, which consists of MRE11–RAD50–NBS1 in a 2:2:1 ratio, is a central hub of DNA double-strand break repair pathways (*Rotheneder et al., 2023*; *Syed and Tainer, 2018*). The MRE11 subunit has nuclease activities that are required for DNA end processing. Several studies have demonstrated that MRE11 physically associates with RPA70N through a motif in the C-terminal region, and that this interaction is required for the correct localization of MRN to replication centers and for the S-phase checkpoint (*Oakley et al., 2009*; *Olson et al., 2007*; *Robison et al., 2004*). We fused the RPA interacting sequence of MRE11 to RPA70N and crystallized the fusion protein in the P212121 space group with one molecule in the asymmetric unit (*Figure 7A and B*, *Figure 1—source data 1*). MRE11 residues (538–563) form a long helix with a short N-terminal loop situated in the basic groove. Positively charged RPA residues R91, R43, R41, R81 and K88 interact with negatively charged D544, D549, E552, D559 and D543 from MRE11. Among these residues, D543 and D544 were previously shown to be important for RPA association (*Olson et al., 2007*). Mutation of the conserved arginine residues in RPA70N (R41, R43, R91) significantly reduced the binding of RPA70N to MRE11 peptide

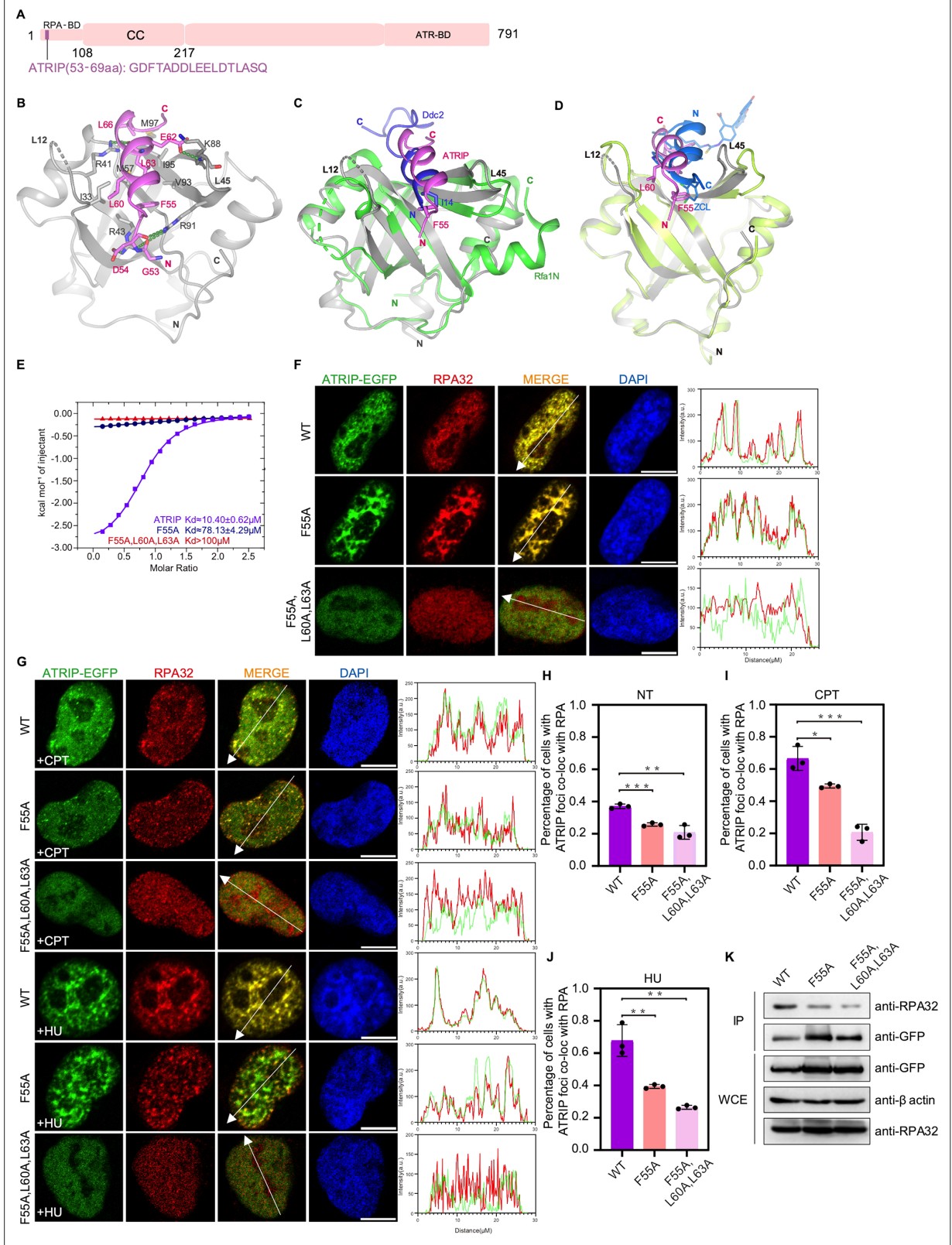

**Figure 6.** Structure of the RPA70N–ATRIP complex. (**A**) Linear domain diagram of ATRIP, showing the position and sequence of the RPA70N-interacting motif. CC, coiled-coiled domain; ATR-BD, ATR binding domain. (**B**) Ribbon representation of the RPA70N–ATRIP crystal structure. The ATRIP peptide is colored in violet and RPA70N in light grey. Important interacting residues are shown as sticks, and green dashed lines indicate hydrogen bonds or salt bridges. (**C**) Alignment of the RPA70N–ATRIP structure with the Ddc2–Rfa1N structure (PDB: 5OMB), showing that ATRIP and Ddc2 bind to RPA70N in

*Figure 6 continued on next page*

*Figure 6 continued*

the same direction. RPA70N–ATRIP components are colored as in (**B**), Ddc2 is colored in green and Rfa1N is colored in blue. (**D**) Superposition of the RPA70N–ATRIP structure with that of the RPA70N-stapled peptide complex (PDB:4NB3). For 4NB3, RPA70N is colored in light-green and the stapled peptide is colored in marine. ZCL is 3,4-dichloro-substituted phenylalanine. The direction of the stapled peptide is reversed when compared to that of ATRIP. (**E**) ITC titration data for WT ATRIP (53–69 aa) and for F55A or F55A/L60A/L63A mutant peptides with RPA70N. (**F, G**) HeLa cells expressing ATRIP–EGFP, ATRIP (F55A)–EGFP, or ATRIP (F55A/L60A/L63A)–EGFP were treated with medium control (NT), CPT (2 µM, 2 h) or HU (2 mM, 3 h), fixed and immunostained with an anti-RPA32 antibody. The scale bar is 10 µm. The intensities of the fluorescent signals for ATRIP–EGFP, ATRIP (F55A)–EGFP or ATRIP (F55A/L60A/L63A)–EGFP and RPA32 are displayed on the right. (**H, I and J**) Quantification of data from (**F**) and (**G**), data are presented as mean ± s.d. of three independent experiments. 100 cells from each experiment were analyzed, and cells containing more than three bright ATRIP and RPA co-localization foci were defined as positive. Statistical analysis was performed using a two-tailed Student's t-test (**** $P<0.0001$, *** $P<0.001$, ** $P<0.01$, * $P<0.05$). (**K**) Immunoprecipitation and western blots showed that mutation of ATRIP residues reduced RPA association. Anti-EGFP magnetic beads were used to carry out immunoprecipitations, which were followed by probing with an anti-RPA32 antibody. IP, immunoprecipitation; WCE, whole cell extract.

The online version of this article includes the following source data and figure supplement(s) for figure 6:

**Source data 1.** Raw data for all of the western blots shown in *Figure 6*.

**Figure supplement 1.** Characterization of the RPA70N–ATRIP interaction (related to *Figure 6*).

(*Figure 7—figure supplement 1A–1C*). In addition to these charge–charge interactions, hydrophobic MRE11 F540, L545, I548, A551 and A555 residues pack against the hydrophobic residues at the bottom of the RPA70N basic groove. In addition, MRE11 N556 and S547 form several hydrogen bonds with RPA70N Q61 and N85. ITC titration showed that MRE11 binds to RPA70N with a $K_D$ of round 16.3 µM, whereas the triple mutation of the hydrophobic residues F540, L545 and I548 resulted in a $K_D$ value that was much higher than 100 µM (*Figure 7C*, *Figure 7—figure supplement 1D and E*). Similar to the observation by *Olson et al., 2007* in fibroblast IMR90 cells, MRE11–FLAG colocalized with endogenous RPA in HeLa cells (*Figure 7D*). MRE11–FLAG with F540/L545/I548 triple mutation led to slightly worse colocalization, in agreement with its reduced binding to RPA in coimmunoprecipitation experiments (*Figure 7D and E*).

## Structure of the RPA70N–RAD9 peptide complex

The RAD9–HUS1–RAD1 (9-1-1) complex is a heterotrimeric ring-shaped molecule that is loaded onto DNA at sites of DNA damage. It plays important roles in the DNA damage-induced checkpoint response (*Doré et al., 2009*; *Parrilla-Castellar et al., 2004*; *Sohn and Cho, 2009*; *Xu et al., 2009*). The RAD9 subunit has an N-terminal domain and a C-terminal domain connected by an inter-domain connecting loop (IDCL). Previous studies showed that RAD9 directly interacts with RPA70N through a CRD motif (checkpoint recruit domain) in the C-terminal domain; mutation of the CRD motif affected RAD9 localization and ATR checkpoint signaling (*Figure 8A*; *Wu et al., 2005*; *Xu et al., 2008b*). We fused the CRD to RPA70N and crystallized the fusion protein in the P3221 space group (*Figure 8A and B*, *Figure 1—source data 1*). The C-terminal part of CRD forms a short two-turn helix, while the N-terminal part adopts an extended conformation. RAD9 D297, D301 and D302 interact with RPA70N R31, R43 and R91 residues, respectively (*Figure 8B*). RAD9 F298 inserts into the side pocket, as observed for HelB and DNA2 (*Figures 8C, 2D and F*). Several hydrophobic residues from RAD9 (I303, M307, I308 and M310) pack with hydrophobic residues at the bottom of the RPA70N groove. ITC experiments showed that RAD9 CRD binds to RPA70N with a $K_D$ of around 20.5 µM, but mutation of F298, M307 and I303 resulted in a substantially higher $K_D$ value (*Figure 8D*, *Figure 8—figure supplement 1A and B*). In untreated HeLa cells, WT RAD9–EGFP generally colocalized with RPA. CPT or HU treatment increased the colocalization even more, while the RAD9 F298/M307/I303 triple mutant had reduced RPA–RAD9 colocalization (*Figure 8E*). This is in agreement with results obtained in U2OS cells by *Xu et al., 2008b*, who mutated the aspartate residues in RAD9 CRD to lysine residues. The RAD9 F298/M307/I303 triple mutant pulled down much less RPA than did WT RAD9 (*Figure 8F*). In addition, mutation of the conserved arginine residues in RPA70N (R31, R41, R43, R91) disrupted the binding of RPA70N to RAD9 peptide (*Figure 8—figure supplement 1C–1F*). Our results, in combination with those of previous studies, lay out the details of RPA70N/RAD9-mediated 9-1-1 clamp complex recruitment to DNA damage sites.

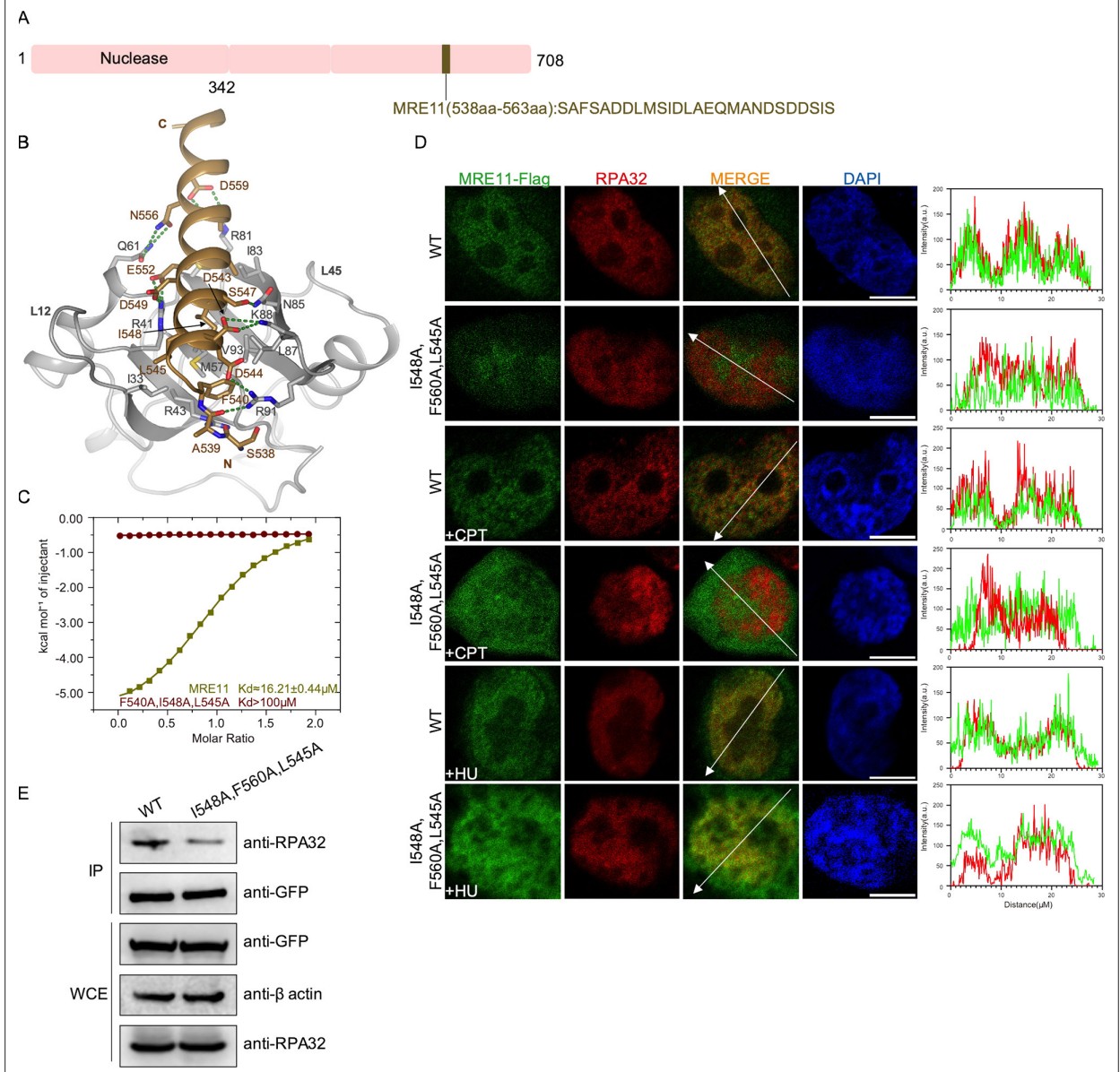

**Figure 7.** Structure of the RPA70N–MRE11 complex. (**A**) Linear domain diagram of MRE11, showing the position and sequence of the RPA70N-interacting motif. (**B**) Ribbon representation of the RPA70N–MRE11 crystal structure. The MRE11 peptide is colored in brown and RPA70N in light grey. Interacting residues are shown as sticks, and green dashed lines indicate hydrogen bonds or salt bridges. (**C**) ITC titration data for WT MRE11 (538–563aa) or F540A/I548A/L545A mutant peptide with RPA70N. (**D**) HeLa cells expressing MRE11–Flag or MRE11–Flag (F540A/I548A/L545A) were treated with medium control, CPT (2 µM, 2 h) or HU (2 mM, 3 h), fixed and immunostained with anti-Flag and anti-RPA32 antibodies. The scale bar is 10 µm. The intensity of fluorescent signals for MRE11–Flag, MRE11–Flag (F540A/I548A/L545A) and RPA32 are displayed on the right. (**E**) Immunoprecipitation and western blot showed that mutation of MRE11 residues reduced RPA association. Anti-EGFP magnetic beads were used to carry out immunoprecipitations, which were followed by probing with an anti-RPA32 antibody. IP, immunoprecipitation; WCE, whole cell extract.

The online version of this article includes the following source data and figure supplement(s) for figure 7:

**Source data 1.** Raw data for all of the western blots shown in *Figure 7*.

**Figure supplement 1.** Characterization of the RPA70N–MRE11 interaction (related to *Figure 7*).

## Structure of the RPA70N–ETAA1 peptide complex

ETAA1 is a newly identified ATR activator that is able to promote restart of stalled replication forks to maintain genome integrity (*Bass et al., 2016*; *Feng et al., 2016*; *Haahr et al., 2016*; *Lee et al., 2016*). ETAA1 is recruited to the DNA damage site via its two RPA-binding motifs (RBM): RBM1 interacts with

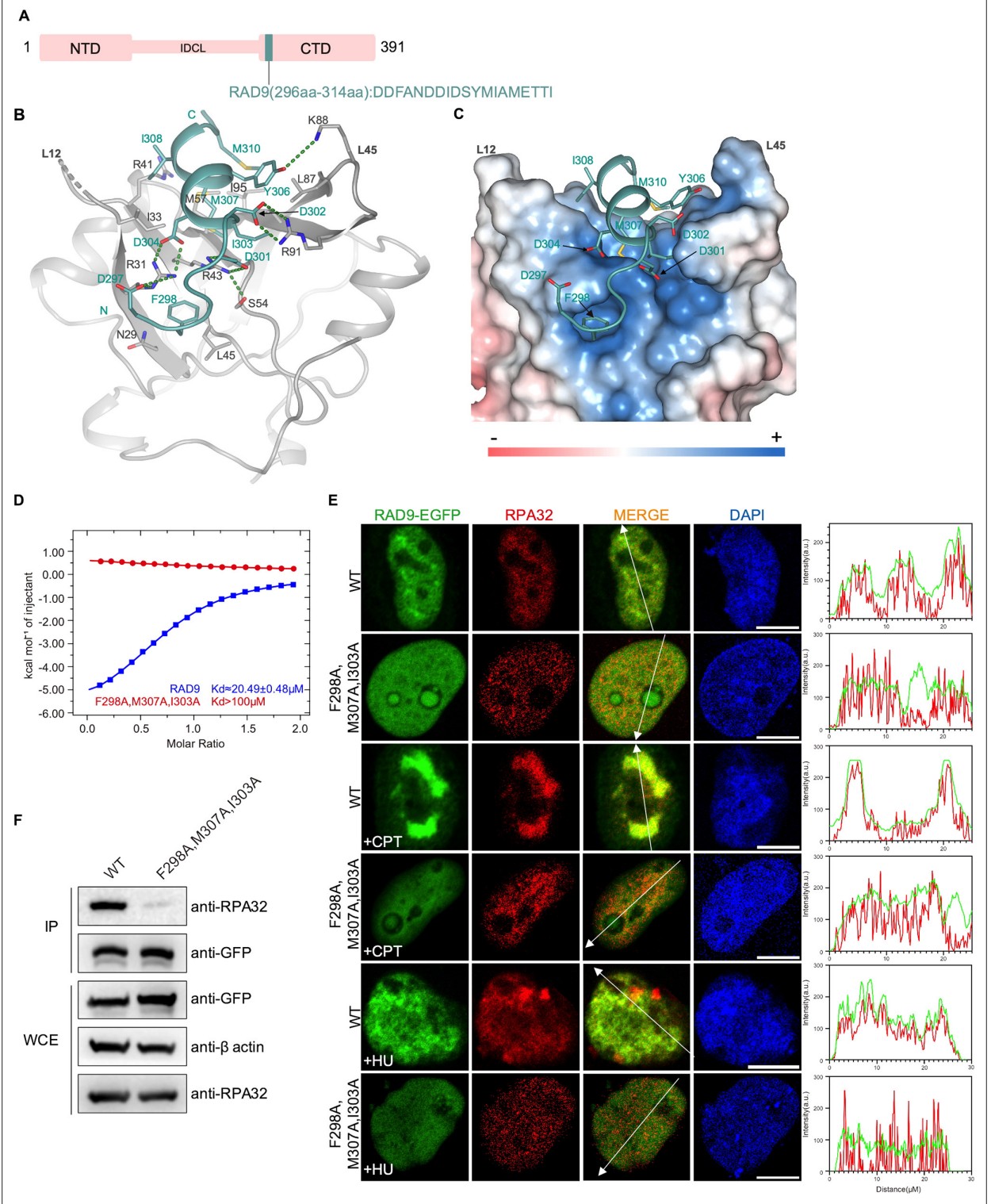

**Figure 8.** Structure of the RPA70N–RAD9 complex. (**A**) Linear domain diagram of RAD9, showing the position and sequence of the RPA70N-interacting motif. IDCL, interdomain connecting loop. (**B**) Ribbon representation of the RPA70N–RAD9 crystal structure. The RAD9 peptide is colored in teal and RPA70N in light grey. Interacting residues are displayed as sticks, and green dashed lines indicate hydrogen bonds or salt bridges. (**C**) Structure of RPA70N in complex with RAD9, showing the surface charge of RPA70N. Hydrophobic and negatively charged residues of RAD9 are displayed as sticks, while the RAD9 peptide is colored in teal. F298 inserts into the side pocket. (**D**) ITC titration data for WT RAD9 (296–314aa) or F298A/M307A/I303A mutant peptide with RPA70N. (**E**) HeLa cells expressing RAD9–EGFP or RAD9 (F298A/M307A/I303A)–EGFP were treated with medium control, CPT (2 μM, 2 h) or HU (2 mM, 3 h), fixed and immunostained with an anti-RPA32 antibody. The scale bar is 10 μm. The intensities of fluorescent signals for

*Figure 8 continued on next page*

*Figure 8 continued*

RAD9–EGFP, RAD9 (F298A/M307A/I303A)–EGFP and RPA32 are displayed on the right. (**F**) Immunoprecipitation and western blots show that mutation of RAD9 residues reduced RPA association. Anti-EGFP magnetic beads were used to carry out immunoprecipitations, which were followed by probing with an anti-RPA32 antibody. IP, immunoprecipitation; WCE, whole cell extract.

The online version of this article includes the following source data and figure supplement(s) for figure 8:

**Source data 1.** Raw data for all of the western blots shown in *Figure 8*.

**Figure supplement 1.** Characterization of the RPA70N–RAD9 interaction (related to *Figure 8*).

RPA70N and RBM2 interacts with RPA32C (*Figure 9A*; *Bass et al., 2016*; *Feng et al., 2016*; *Haahr et al., 2016*; *Lee et al., 2016*). We fused RBM1 to RPA70N and crystallized the fusion protein in the P3221 space group (*Figure 9A and B*, *Figure 1—source data 1*). RBM1 forms a three-turn α helix with a short N-terminal extension and sits in the basic groove (*Figure 9B*). ETAA1 D606 and D607 interact with RPA70N R31, R43 and R91. ETAA1 W600, L609, L610, Y611 and A613 pack with hydrophobic residues (I33, M57, L87, V93 and I95) from the RPA70N basic groove (*Figure 9B*). In addition, W600 and Y611 stack with the guanidino groups from R91 and R41, respectively, to form cation-π interactions (*Figure 9B*). ITC titration showed that the affinity between ETAA1 RBM1 and RPA70N was relatively high among peptides tested in this study, with a $K_D$ of around 3.9 μM (*Figure 9C* and *Figure 9—figure supplement 1A*). ETAA1 W600A/L610A/Y611A triple mutation greatly reduced the affinity between ETAA1 RBM1 and RPA70N (*Figure 9C* and *Figure 9—figure supplement 1B*). In HeLa cells, ETAA1 The W600A/L610A/Y611A triple mutant produced significantly reduced numbers of EGFP–ETAA1 and RPA foci, but triple mutant ETAA1 and RPA still have a substantial amount of colocalization (*Figure 9D–G*). This is in agreement with previous studies showing that ETAA1 has two RPA-binding motifs and that mutated RBM1 alone couldn't completely abolish ETAA1–RPA association (*Bass et al., 2016*; *Feng et al., 2016*; *Haahr et al., 2016*; *Lee et al., 2016*). Nonetheless, our results and findings from other groups all suggest that RBM1 is required for the normal function of ETAA1 in the DNA damage response (*Figure 9D–G*; *Bass et al., 2016*; *Feng et al., 2016*; *Haahr et al., 2016*; *Lee et al., 2016*). Co-IP experiments showed reduced binding of the EGFP–ETAA1 W600A/L610A/Y611A triple mutant to RPA, and that mutation of the conserved arginine residues in RPA70N (R31, R41, R43, R91) disrupted the binding of RPA70N to the ETAA1 peptide (*Figure 9H*, *Figure 9—figure supplement 1C–1F*). These data further confirmed the RPA70N–ETAA1 RBM1 interacting residues uncovered in the structure. Our results provide a structural basis for RPA70N-mediated RPA–ETAA1 interaction.

## Discussion

It is well established that RPA plays important roles in DNA replication, recombination and repair (*Caldwell and Spies, 2020*; *Chen and Wold, 2014*; *Fanning et al., 2006*; *Iftode et al., 1999*; *Maréchal and Zou, 2015*; *Wold, 1997*; *Zou et al., 2006*). Many RPA–protein interactions are mediated by the flexibly tethered RPA70N domain. However, RPA70N–partner interactions are often weak and highly dynamic (*Caldwell and Spies, 2020*; *Fanning et al., 2006*). As a result, high-resolution structures of RPA70N bound to partner peptides are rare relative to the large number of proteins that RPA70N interacts with. To overcome this problem, inspired by the fusion approach first employed by *Bochkareva et al., 2005* to solve the RPA70N–p53 complex structure, we systematically screened RPA70N–partner fusion constructs for crystallization and determined nine complex structures of proteins involved in the DNA damage response (*Figure 10A*, *Figure 1F–N*, *Figure 1—source data 1*). Superposition of the nine structures determined in this study with apo RPA70N showed that most of the Cα atoms of RPA70N were at nearly identical positions, with RMSD values smaller than 0.3 Å (*Figure 10B*). The L12 and L45 region displayed small conformational changes to accommodate different peptides. Overall, it appears that the interaction of RPA70N with partner protein motifs relies on the movement of side chains of the conserved positively charged and hydrophobic residues. Not surprisingly, mutation of the positively charged residues in RPA70N weakened or nearly abolished RPA70N's ability to bind target peptides (*Figure 2—figure supplement 1G*, *Figure 3—figure supplement 2C and F*, *Figure 4—figure supplement 1D and E*, *Figure 5—figure supplement 1E and F*, *Figure 6—figure supplement 1D*, *Figure 7—figure supplement 1A*, *Figure 8—figure supplement 1C*, *Figure 9—figure supplement 1C*). RPA70 R31H or R31C mutation is found in some cancer patients (from the COSMIC database), which might be related to its role in protein–protein interaction. The newly solved

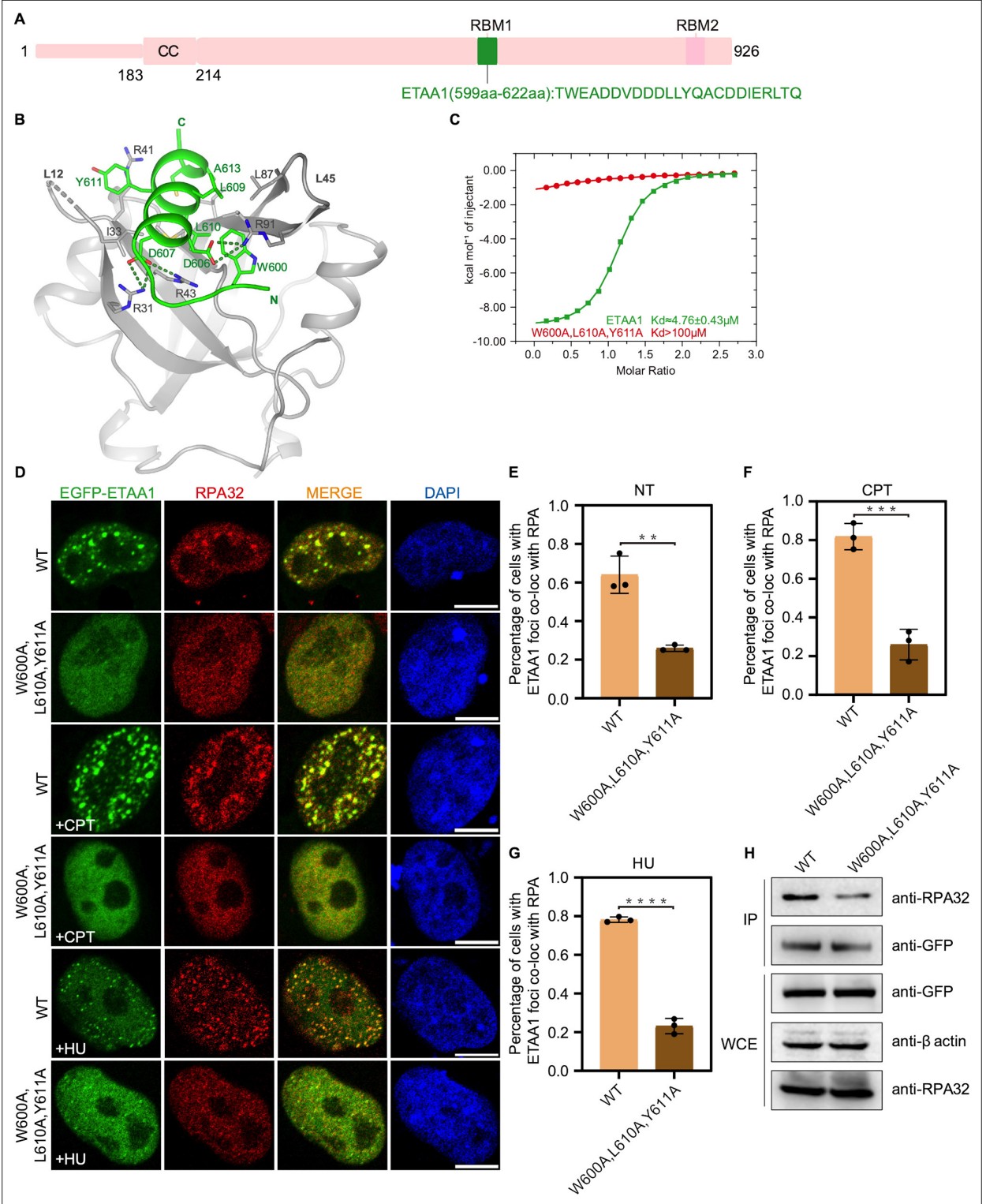

**Figure 9.** Structure of the RPA70N–ETAA1 complex. (**A**) Linear domain diagram of ETAA1, showing the position and sequence of the RPA70N-interacting motif. ETAA1 has a coiled-coiled domain (CC) and two RPA-binding motifs (RBM1, RBM2). (**B**) Ribbon representation of the RPA70N–ETAA1 crystal structure. ETAA1 peptide is colored in bright green and RPA70N in light grey. Interface residues are shown as sticks, and green dashed lines indicate hydrogen bonds or salt bridges. (**C**) ITC titration data for the WT ETAA1 peptide (599–622 aa) or for the W600A/L610A/Y611A mutant peptide with RPA70N. (**D**) HeLa cells expressing EGFP–ETAA1 or EGFP–ETAA1(W600A/L610A/Y611A) were treated with medium control (NT), CPT (2 µM, 2 h) or HU (2 mM, 3 h), fixed and immunostained with an anti-RPA32 antibody. The scale bar is 10 µm. (**E–G**) Quantification of data from (**D**), data are presented

*Figure 9 continued on next page*

*Figure 9 continued*

as mean ±s.d. of three independent experiments. 100 cells from each experiment were analyzed, and cells containing more than three bright ETAA1 and RPA co-localization foci were defined as positive. Statistical analysis was performed using a two-tailed Student's t-test (**** $P<0.0001$, *** $P<0.001$, ** $P<0.01$, * $P<0.05$). (**H**) Immunoprecipitation and western blotting showed that mutation of ETAA1 residues reduced RPA association. Anti-EGFP magnetic beads were used to carry out immunoprecipitations, which were followed by probing with an anti-RPA32 antibody. IP, immunoprecipitation; WCE, whole cell extract.

The online version of this article includes the following source data and figure supplement(s) for figure 9:

**Source data 1.** Raw data for all of the western blots shown in *Figure 9*.

**Figure supplement 1.** Characterization of the RPA70N–ETAA1 interaction (related to *Figure 9*).

structures also confirmed previous findings that RPA70N binds to a partner sequence through two interfaces:the basic and hydrophobic groove; and the side pocket, which is also basic and hydrophobic. The side pocket is not always used, as seen in the data for BLMp2, ATRIP, MRE11, ETAA1 and PrimPol (*Figure 10A*). In theory, the empty side pocket could be the binding site of another peptide. This second peptide could be a not-yet-identified sequence in the protein RPA70N bound to or from another molecule. More importantly, we found that RPA70N is able to coordinate peptide binding to its two interfaces through diverse means, such as inverted direction, rotation/tilt of the bound helix, kinked conformation, or dimerization (*Figure 10A*). The versatile interaction processes are presumably customized to the different protein sequences that RPA encounters. One could imagine that RPA70N must be able to recruit different partners under different scenarios.

Of particular interest is that many of the partner peptides appear to be able to connect two RPA70N domains (*Figure 10A*). If we expand the observed dimer, we could get a string of RPA70N domains connected by BLMp1, BLMp2 or RMI1. Intriguingly, some of these partner proteins (BTR complex, WRN, ATRIP, p53 and ETAA1) are themselves often dimers or oligomers (*Cho et al., 1994*; *Compton et al., 2008*; *Deshpande et al., 2017*; *Hodson et al., 2022*; *Thada and Cortez, 2021*). Even for RPA70N-interacting proteins that are not dimers or oligomers, they often associate with each other directly or indirectly. For example, RPA is able to recruit ATR/ATRIP to stalled replication forks, which nucleates many RPA70N-interacting DNA damage response proteins (*Figure 10—figure supplement 1A*; *Bass et al., 2016*; *Feng et al., 2016*; *Haahr et al., 2016*; *Lee et al., 2016*; *Maréchal and Zou, 2015*; *Saldivar et al., 2017*). *Frattini et al., 2021* proposed the formation of a nucleation complex including RPA–ssDNA, ATR/ATRIP, TopBP1, MRN and the 9-1-1 complex, which stabilizes TopBP1 at stalled replication forks. TopBp1 in turn condensates to dynamic higher-order assemblies by multivalent cooperative interactions to achieve robust ATR activation and signal amplification (*Frattini et al., 2021*). So there are indeed multiple copies of partner peptides in close range. One RPA usually covers around 30 nt ssDNA (*Kim et al., 1994*); for medium to long ssDNA, there are multiple copies of bound RPA. The bridge-forming nature of some of the partner peptides in combination with many RPAs on ssDNA could greatly enhance the efficiency of partner recruitment when needed for DNA damage response (*Figure 10—figure supplement 1B and C*).

Multivalent interaction processes could also serve as an intrinsic layer of regulation in addition to signal transduction pathways such as protein modifications. Under normal conditions, RPA molecules are not clustered on ssDNA and have a relatively weak affinity for many DNA damage response proteins, as shown by the dissociation constant values measured in this study and previous studies (*Hegnauer et al., 2012*; *Lee et al., 2018*; *Souza-Fagundes et al., 2012*; *Yeom et al., 2019*). Upon DNA damage, RPAs nucleate on exposed ssDNA quickly, owing to their sub-nanomolar affinity for ssDNA, and recruit corresponding proteins. With sufficient copies of RPA bound to ssDNA, the weak affinity of monomeric RPA70N toward target protein is now overwhelmed by multiple interaction interfaces (*Figure 10—figure supplement 1B and C*). A recent study showed that RPA has a strong propensity to assemble into dynamic condensates (undergo phase separation), which is likely to be driven by RPA2 and could be stimulated by ssDNA binding (*Spegg et al., 2023*). More importantly, the data demonstrate that RPA condensation enhances interactions with the BTR complex. The multivalent interactions that we observed in the crystal structures could contribute significantly to these condensation-driven DNA damage response processes (*Figure 10—figure supplement 1B and C*).

In our imaging analysis, HelB, BLM, WRN, ATRIP and ETAA1 formed a substantial number of foci and colocalized with endogenous RPA (stained with antibodies against RPA2), and this colocalization was further stimulated by CPT or HU treatment (*Figure 2H-J*, *Figure 3J-M*, *Figure 5H-K*, *Figure 6F-J*,

**A**

| | Protein | Basic groove | Side pocket | Direction | Bridging two RPA70N in crystal | PDB code |
|---|---|---|---|---|---|---|
| This study | HelB | Yes | Yes | | No | 7XUT |
| | BLMp2 | Yes | No | | Yes | 7XUV |
| | BLMp1 | Yes | Yes | | Yes | 7XV0 |
| | RMI1 | Yes | Yes | | Yes | 7XV1 |
| | WRN | Yes | Yes | | Yes | 7XV4 |
| | ATRIP | Yes | No | | No | 7XUW |
| | MRE11 | Yes | No | | No | 8JZY |
| | RAD9 | Yes | Yes | | No | 8K00 |
| | ETAA1 | Yes | No | | No | 8JZV |
| Previous studies | p53 | Yes | Yes | | Yes | 2B3G |
| | DNA2 | Yes | Yes | | No | 5EAY |
| | PrimPol | Yes | No | | No | 5N85 |
| | Ddc2 | Yes | No | | No | 5OMB |

**B**

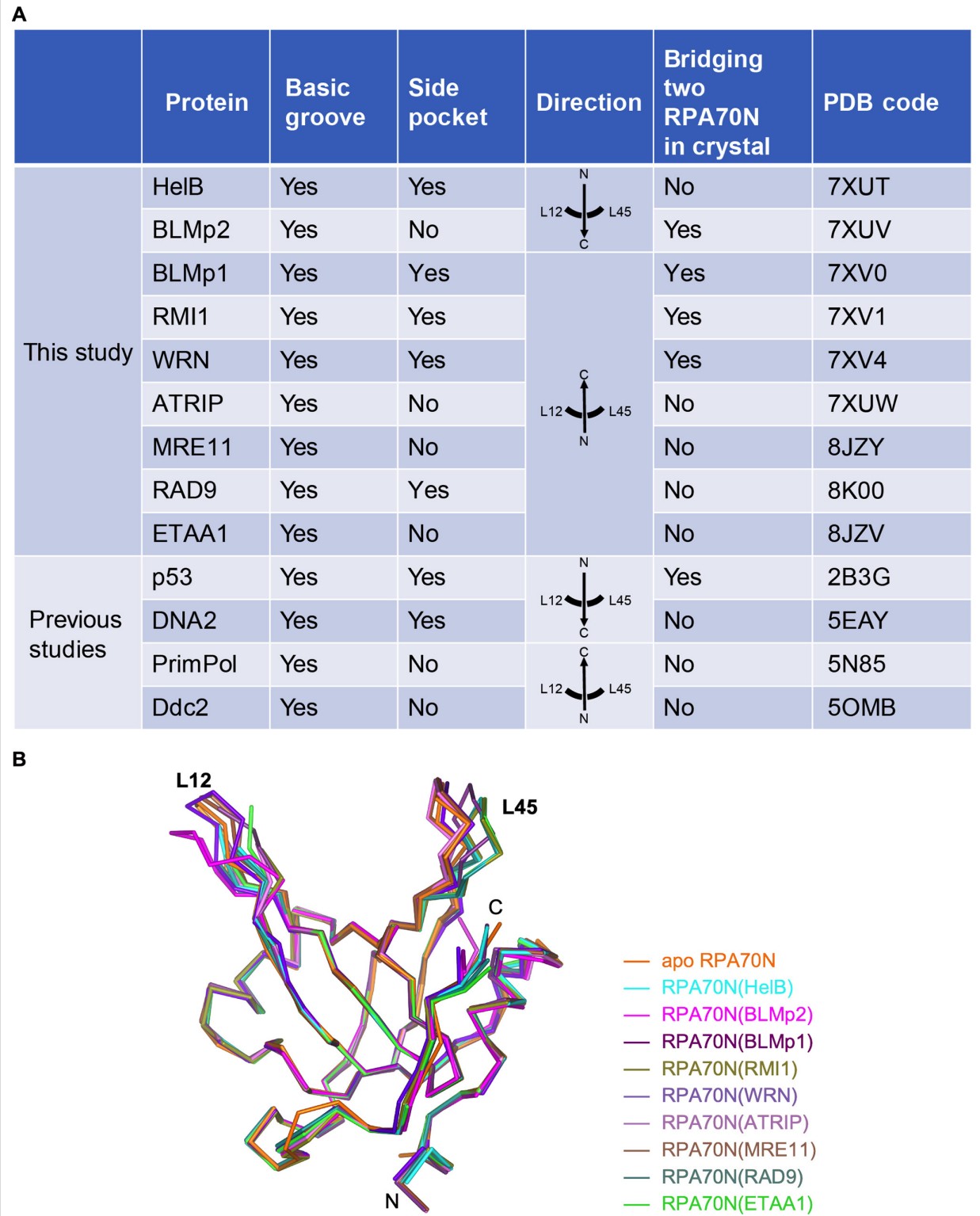

Figure 10. Comparison of RPA70N–peptide complex structures. (**A**) Summary of the RPA70N-binding proteins for which RPA70N complex structures are available (Ddc2 binds to Rfa1N). (**B**) Superposition of apo RPA70N (orange, PDB: 2B29) with RPA70N structures determined in this study, colored using the peptide color code used in *Figure 1F–N*.

The online version of this article includes the following source data and figure supplement(s) for figure 10:

**Source data 1.** Editable word file of *Figure 10A*.

**Figure supplement 1.** Model of protein recruitment in the RPA70N-mediated DNA damage response (related to *Figure 10*).

*Figure 9D-G*). Mutation of interface residues significantly weakened the formation of foci and colocalization (*Figure 2H-J*, *Figure 3J-M*, *Figure 5H-K*, *Figure 6F-J*, *Figure 9D-G*). RMI1, MRE11 and RAD9 didn't form distinct foci, but still colocalized with RPA to different degrees, a response that was also weakened by interface mutations (*Figures 4G, 7D and 8E*). Thus, it's possible that ssDNA enriches RPA at the damage site, which promotes the association of RPA with partner proteins. At the same time, RPA70N binding to partner proteins with multivalent sites in turn promotes RPA oligomeric assembly.

In summary, the structural snapshots and biochemical analyses that we present here shed light on the diverse modes of RPA70N interacting with DNA damage response proteins. These interactions could serve to increase the avidity of RPA70N binding. Our results have provide a molecular basis for partner protein recruitment by RPA70N. Further studies with full-length RPA and RPA-interacting proteins are required to delineate the complex interaction network of RPA in DNA damage response.

# Materials and methods

## Key resources table

| Reagent type (species) or resource | Designation | Source or reference | Identifiers | Additional information |
|---|---|---|---|---|
| Strain, strain background (*Escherichia coli*) | Rosetta 2(DE3) chemically competent cell | Novagen | 71402 | |
| Strain, strain background (*E. coli*) | *E. coli* BL21(DE3) cells | Novagen | 69450 | |
| Strain, strain background (*E. coli*) | Trelief 5α chemically competent cell | TSINGKE | TSC-C01-100 | |
| Cell line (*Homo sapiens*) | HEK293T | ATCC | CRL-3216 | Authenticated by STR profiling, no mycoplasma contamination |
| Cell line (*H. sapiens*) | HeLa | Cell bank of the Chinese Academy of Sciences | SCSP-504 | Authenticated by STR profiling, no mycoplasma contamination |
| Antibody | Anti-GFP rabbit monoclonal antibody | Beyotime | AF1483 | WB (1:5000) |
| Antibody | Anti-RPA32 rabbit monoclonal antibody | Beyotime | AG3115 | WB (1:2000), IF (1:50) |
| Antibody | Anti-β-actin mouse monoclonal antibody | Proteintech | 66009–1-Ig | WB (1:5000) |
| Antibody | Anti-flag tag mouse monoclonal antibody | Beyotime | AF2852 | IF (1:50) |
| Antibody | Horseradish peroxidase labelled goat anti-mouse polyclonal secondary antibody | Beyotime | A0216 | WB (1:5000) |
| Antibody | Horseradish peroxidase labelled goat anti-rabbit polyclonal secondary antibody | Beyotime | A0208 | WB (1:5000) |
| Antibody | Alexa 568 goat anti-rabbit polyclonal antibody IgG (H+L) cross-adsorbed secondary antibody | Thermo Fisher Scientific | A-11036 | IF (1:400) |
| Antibody | Alexa 488 goat anti-mouse polyclonal antibody IgG (H+L) cross-adsorbed secondary antibody | Thermo Fisher Scientific | A32723 | IF (1:400) |
| Recombinant DNA reagent | pcDNA3.1(+) | Invitrogen | V79020 | |
| Recombinant DNA reagent | pRSFDuet-1 vector | Novagen | 71341 | |
| Commercial assay or kit | ClonExpress II One Step Cloning Kit | Vazyme | C112-01/02 | |
| Chemical compound, drug | Hydroxyurea | Beyotime | Cat#S1961 | |
| Chemical compound, drug | (S)-(+)-Camptothecin, 98% | J&K | Cat#7689-03-4 | |
| Chemical compound, drug | 2-(4-Amidinophenyl)–6-indolecarbamidine dihydrochloride | Beyotime | C1006 | DAPI staining solution used to stain nucleus |

## Cloning, protein expression and purification

The DNA sequences of human RPA70N (residues 1–120), HelB (residues 496–519), BLMp1 (residues 146–165), BLMp2 (residues 550–570), RMI1 (residues 243–262), WRN (residues 435–451), ATRIP (residues 53–69), MRE11 (residues 538–563), RAD9 (residues 296–314) and ETAA1 (residues 599–622) were individually cloned into a modified pRSFDuet-1 vector (Novagen). This vector fuses an N-terminal 6-His-sumo tag to the target gene using ClonExpress II One Step Cloning Kit (Vazyme). RPA70N–peptide fusion constructs were cloned into the same expression vector. RPA70N, RPA70N–peptide fusion proteins and all peptides were expressed and purified with similar steps. The recombinant plasmids were transformed into *E. coli* BL21(DE3) cells or Rosetta 2 (DE3) cells (Novagen), which were grown in LB medium at 37 °C until the OD 600 reached 0.6–0.8. Overexpression of proteins was induced by the addition of 0.5 mM isopropyl β-D-thiogalactopyranoside (IPTG), followed by incubation at 20 °C for 14 h. Cells were harvested by centrifugation, resuspended in lysis buffer (20 mM Tris-HCl, 200 mM NaCl, 20 mM imdazole, 10% glycerol, 0.3 mM TCEP, pH 8.0), and lysed by a high-pressure homogenizer at 4 °C. The cell lysate was centrifuged at 12,000 rpm for 40 min to obtain soluble extract. After nickel affinity pull-down, 6-His-sumo tag was cleaved off by Ulp1 protease and removed by a second nickel column. Flow-through was then passed through a Source 15Q column (Cytiva) and eluted with a gradient of 0–1 M NaCl in a buffer of 20 mM Tris-HCl, pH 8.0, 10% glycerol, 0.3 mM TCEP. Fractions containing target proteins were pooled and concentrated, then further purified on a Superdex 75 increase gel filtration column (Cytiva) in a buffer containing 20 mM Tris-HCl, pH 8.0, 150 mM NaCl, 0.3 mM TCEP. The purified RPA70N–peptide fusion proteins were concentrated to around 20–25 mg/ml for crystallization. RPA70N and all peptides were concentrated to suitable concentrations for ITC titrations.

## Crystallization

For all of the RPA70N–peptide fusion proteins, crystallization screenings were performed using 96-well plates in a sitting drop mode at 4°C. The RPA70N–HelB fusion protein crystallized in 20% (w/v) PEG 3350, 200 mM calcium chloride. The RPA70N-BLMp1 fusion protein crystallized in 100 mM sodium citrate pH 5.6, 2000 mM ammonium sulfate, 200 mM potassium/sodium tartrate. RPA70N–BLMp2 fusion protein crystallized in 100 mM sodium acetate pH 4.6, 8% (w/v) PEG 4000. The RPA70–WRN fusion protein crystallized in 20% (w/v) PEG 3350, 200 mM ammonium sulfate. The RPA70N–RMI1 fusion protein crystallized in 100 mM sodium acetate pH 4.6, 30% (w/v) PEG 2000 MME, 200 mM ammonium sulfate. The RPA70N–ATRIP fusion protein crystallized in 100 mM Tris-HCl pH 8.5, 2400 mM ammonium sulfate. The RPA70N–MRE11 fusion protein crystallized in 100 mM HEPES pH 7.5, 1260 mM ammonium sulfate. The RPA70N–RAD9 fusion protein crystallized in 100 mM sodium acetate pH 4.6, 25% (w/v) PEG 4000, 200 mM ammonium sulfate. The RPA70N–ETAA1 fusion protein crystallized in 20% (w/v) PEG 3350, 200 mM ammonium chloride. Crystals were cryo-protected in their respective well solutions supplemented with 20% ethylene glycol and flash-frozen in liquid nitrogen.

## Structure determination and refinement

Diffraction data were collected at Beamline stations BL17U1, BL18U1 and BL19U1 at Shanghai Synchrotron Radiation Facility (SSRF, Shanghai, China). The data were integrated and scaled using XDS, the CCP4 program Pointless and Aimless (*Evans and Murshudov, 2013*; *Kabsch, 2010*; *Winn et al., 2011*). The structures of RPA70N–peptide fusion constructs were determined by molecular replacement using the RPA70N structure from PDB 5EAY as an initial searching model with Phaser (*McCoy et al., 2007*). The structural model was built using Coot (*Emsley and Cowtan, 2004*) and refined using PHENIX (*Liebschner et al., 2019*). Figures were generated using PyMOL (The PyMOL Molecular Graphics System, Version 2.0 Schrödinger, LLC). The statistics for the data collection and refinement are shown in *Figure 1—source data 1*.

## Isothermal titration calorimetry (ITC)

All ITC titrations were carried out using a MicroCal PEAQ-ITC instrument (Malvern) at 25 °C with different peptides in the syringe and RPA70N in the cell. RPA70N and peptide samples were dialyzed against a working buffer consisting of 20 mM HEPES, 100 mM NaCl, 1 mM DTT, pH 7.5. Each titration was carried out with 19 injections, spaced 150 s apart, stir speed at 500 rpm. The acquired calorimetric titration data were analyzed with Origin 7.0 software using the 'One Set of Binding Sites' fitting model.

## Cell culture

HeLa cells (SCSP-504, Cell bank of the Chinese Academy of Sciences) and HEK293T (CRL-3216, ATCC) cells used in this study were authenticated by STR (short tandem repeat) profiling and tested negative for mycoplasma contamination. They were cultured at 37 °C with 5% $CO_2$ in Dulbecco's modified Eagle's medium containing 10% fetal bovine serum (ExCell Bio), 100 units/ml penicillin, and 100 µg/ml streptomycin. HeLa cells and HEK293T cells were seeded into glass-bottom dishes and cultured overnight. The next day, 1–2 µg of each plasmid and 3–6 µL of PEI reagent (Polyscience) were mixed and transferred to each well for transient transfection.

## Antibodies

The following antibodies were used in this study: anti-GFP rabbit monoclonal antibody (Beyotime, AF1483), anti-RPA32 rabbit monoclonal antibody (Beyotime, AG3115), anti-FLAG tag mouse monoclonal antibody (Beyotime, AF2852), anti-β-actin monoclonal antibody (Proteintech, 66009–1-Ig), goat anti-rabbit polyclonal secondary antibody IgG (Beyotime, A0208), goat anti-mouse polyclonal secondary antibody IgG (Beyotime, A0216), Alexa 568 goat anti-rabbit polyclonal IgG (H+L) cross-adsorbed secondary antibody (Thermo, A_11036), Alexa 488 goat anti-mouse polyclonal IgG (H+L) cross-adsorbed secondary antibody (Thermo, A32723).

## Immunofluorescence

HeLa cells were transiently transfected with HelB–EGFP, EGFP–BLM, EGFP–RMI1, WRN–EGFP, ATRIP–EGFP, MRE11–FLAG, RAD9–EGFP, or EGFP–ETAA1 plasmids (pcDNA3.1(+)). The day after transfection, cells were treated with hydroxyurea (HU) or camptothecin (CPT) for 2–3 hours. Then the cells were washed with phosphate-buffered saline (PBS), fixed with 4% (w/v) paraformaldehyde for 10 min at room temperature, and blocked with QuickBlock blocking buffer (Beyotime). Fixed cells were incubated overnight with an anti-RPA32 antibody (Beyotime) diluted 1:200 in blocking buffer at 4 °C. Then, cells were incubated for 1 h at room temperature in the dark with Alexa Fluor 488 anti-mouse or 568 anti-rabbit secondary antibodies diluted 1:400 in blocking buffer. After that, cells were stained with DAPI staining solution (Beyotime) for 5 min. Finally, they were mounted in a fluorescence quenching solution and imaged on a Nikon C2 confocal microscope. The images of cells excited with individual fluorescent channels were taken separately and merged afterward with Nikon imaging software. For MRE11-FLAG, an anti-FLAG antibody (Beyotime) was also used for immunostaining.

## Immunoprecipitations

DNA sequences of HelB–EGFP, EGFP–BLM, EGFP–RMI1, EGFP–WRN, EGFP–ATRIP, EGFP–ETAA1, RAD9–EGFP, or EGFP–MRE11 were cloned into the plasmid pcDNA3.1(+). 48 h after transfection, HEK 293T cells were washed with phosphate-buffered saline (PBS) and lysed in IP buffer comprising 20 mM Tris-HCl pH 8.0, 150 mM NaCl, 0.5% NP-40, 5% w/v glycerol, ultraNuclease (Yeasen), and 1×PMSF and protease inhibitor cocktail (Topscience). For IP reactions, cleared cell lysates were incubated with anti-GFP magnetic beads (Beyotime) for 4 h at 4 °C with rotation. The beads were washed three times with tris buffered saline (TBS) using a magnetic separator. The bound proteins were eluted with 50 µl 5×SDS-loading buffer. Samples were boiled at 95 °C for 5 min and separated with 10% SDS-PAGE gels for immunoblot analysis.

## Western blotting

Samples were run on 10% SDS-PAGE gels and transferred in a tris-glycine transfer buffer containing 20% methanol and 0.01% SDS onto PVDF membranes. Membranes were blocked using 5% BSA in TBST and incubated with primary antibodies overnight at 4 °C with 5% BSA in TBST. After three washes with TBST, membranes were incubated for 1 h at room temperature with secondary antibodies. Afterward, membranes were washed three times in TBST and imaged (Azure Biosystem C400).

## Acknowledgements

We thank the staff at SSRF BL19U1, BL17U1 and BL18U1 for the collection of X-ray diffraction data. We thank the staff at the National Facility for Protein Science in Shanghai (NFPS) and Westlake University Biomedical Research Core Facilities for assistance with ITC experiments. We thank Dr. Nikola

Pavletich for helpful discussions. We thank Dr. Jie Sun, Dr. Ying Gu, Dr. Yuyuan Zheng, Dr. Panyu Fei and members of Zhou Lab for their kind help. This work was supported by the National Natural Science Foundation of China (31971125 to CZ).

## Additional information

### Funding

| Funder | Grant reference number | Author |
|---|---|---|
| National Natural Science Foundation of China | 31971125 | Chun Zhou |

The funders had no role in study design, data collection and interpretation, or the decision to submit the work for publication.

### Author contributions

Yeyao Wu, Wangmi Fu, Data curation, Formal analysis, Validation, Investigation, Visualization, Writing - original draft, Writing - review and editing; Ning Zang, Investigation; Chun Zhou, Conceptualization, Resources, Data curation, Formal analysis, Supervision, Funding acquisition, Validation, Investigation, Visualization, Methodology, Writing - original draft, Writing - review and editing

### Author ORCIDs

Yeyao Wu http://orcid.org/0000-0003-1144-8508
Chun Zhou http://orcid.org/0000-0002-9257-468X

### Decision letter and Author response

Decision letter https://doi.org/10.7554/eLife.81639.sa1
Author response https://doi.org/10.7554/eLife.81639.sa2

## Additional files

### Supplementary files

- MDAR checklist

### Data availability

Atomic coordinates and structure factors for the reported crystal structures have been deposited with the Protein Data Bank under accession number: 7XUT, 7XUV, 7XUW, 7XV0, 7XV1, 7XV4, 8JZV, 8JZY, 8K00.

The following datasets were generated:

| Author(s) | Year | Dataset title | Dataset URL | Database and Identifier |
|---|---|---|---|---|
| Wu Y, Fu W, Zang N, Zhou C | 2022 | Crystal structure of RPA70N-WRN fusion | https://doi.org/10.2210/pdb7xut/pdb | Worldwide Protein Data Bank, 10.2210/pdb7xut/pdb |
| Wu Y, Fu W, Zang N, Zhou C | 2022 | Crystal structure of RPA70N-RMI1 fusion | https://doi.org/10.2210/pdb7xuv/pdb | Worldwide Protein Data Bank, 10.2210/pdb7xuv/pdb |
| Wu Y, Fu W, Zang N, Zhou C | 2022 | Crystal structure of RPA70N-BLMp2 fusion | https://doi.org/10.2210/pdb7xuw/pdb | Worldwide Protein Data Bank, 10.2210/pdb7xuw/pdb |
| Wu Y, Fu W, Zang N, Zhou C | 2022 | Crystal structure of RPA70N-BLMp1 fusion | https://doi.org/10.2210/pdb7xv0/pdb | Worldwide Protein Data Bank, 10.2210/pdb7xv0/pdb |

*Continued on next page*

*Continued*

| Author(s) | Year | Dataset title | Dataset URL | Database and Identifier |
|---|---|---|---|---|
| Wu Y, Fu W, Zang N, Zhou C | 2022 | Crystal structure of RPA70N-HelB fusion | https://doi.org/10.2210/pdb7xv1/pdb | Worldwide Protein Data Bank, 10.2210/pdb7xv1/pdb |
| Wu Y, Fu W, Zang N, Zhou C | 2022 | Crystal structure of RPA70N-ATRIP fusion | https://doi.org/10.2210/pdb7xv4/pdb | Worldwide Protein Data Bank, 10.2210/pdb7xv4/pdb |
| Wu Y, Fu W, Zang N, Zhou C | 2023 | RPA70N_ETAA1_8JZV | https://doi.org/10.2210/pdb8jzv/pdb | Worldwide Protein Data Bank, 10.2210/pdb8jzv/pdb |
| Wu Y, Fu W, Zang N, Zhou C | 2023 | RPA70N_RAD9_8JZY | https://doi.org/10.2210/pdb8jzy/pdb | Worldwide Protein Data Bank, 10.2210/pdb8jzy/pdb |
| Wu Y, Fu W, Zang N, Zhou C | 2023 | RPA70N_MRE11_ 8K00 | https://doi.org/10.2210/pdb8k00/pdb | Worldwide Protein Data Bank, 10.2210/pdb8k00/pdb |

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
