## [Editor Report]

This important paper advances our understanding of how a eukaryotic single-stranded DNA binding protein, Replication protein-A (RPA) interacts with multiple proteins in DNA transactions. The author provided compelling structure information on an OB-fold called RPA70N (or DBD-F) with 8 different peptides from various DNA metabolisms, which is complemented by in vivo studies. This paper will be of interest to researchers in DNA replication, recombination, and repair as well as structural biologists interested in a weak protein-protein interaction.

---

## [Decision Letter]

**Decision letter after peer review:**

Thank you for submitting your article "Structural characterization of human RPA70N association with DNA damage response proteins" for consideration by *eLife*. Your article has been reviewed by 3 peer reviewers, including Akira Shinohara as Reviewing Editor and Reviewer #1, and the evaluation has been overseen by Volker Dötsch as the Senior Editor.

Based on reviews and discussions with reviewers, we do appreciate that your structural and biochemical results are very important to understand how RPA binds to more than 30 different proteins involved in DNA transactions and applaud the breadth of the work and the high quality of the structures. However, we are very sorry that we can not be more positive about publishing the current version of the paper. Mainly, as you acknowledge, in most of your results, similar structural studies on RPA70N to other proteins (such as p53, ATRIP/Ddc2, PrimPol, DNA2) have been reported. The results in the current version constitute only a limited advance in our understanding of how RPA binds to different binding partners.

On the other hand, if you can provide direct in vitro evidence by biochemical analyses using full-length RPA and a full-length BLM or RMI1 that an RPA-binding protein promotes or changes the oligomerization of RPA in solution or on ssDNA, the paper would provide new insights into RPA70N-interaction with other proteins.

The text would also have to be re-written substantially, so that it flows more logically and through referring to other publications and revising figures.

Essential revisions:

Please determine the interaction between both RPA and RPA-binding proteins by using full-length wild-type and mutant proteins in vitro. Given that BLMp1, BLMp2 and RMI1p could induce the oligomer formation (dimerization) of RPAs, it would be nice to focus on the binding of either BLM or RMI1 to RPA. Importantly, it is highly recommended to quantify a molar ratio of BLM/RMI1 to RPA-the authors may use gel-filtration (you may need the cross-linking to stabilize the binding) or quantitative pull-down assays or others such as low-resolution cryoEM analysis.

1. We do not ask you to do in vivo experiments (as pointed out by Reviewers #1 and #3). However, it is very important to search for literature for in vivo studies to complement your structural studeis and to refer to them in the text in an organized manner.

2. As reviewers #2 and #3 pointed out, you need to add more explanation of your experiment design and results and to discuss your findings logically. And also, you need to reformat your figures by showing RPA-peptides interaction in the same way (see comments by #3 reviewer).

3. Reviewer #3 provided a good example of how to show images of your 3D structure in your Figures. Please use it during your revision.

*Reviewer #1 (Recommendations for the authors):*

The conclusion in the paper is well supported by the results described in the paper. However, given the binding affinity of RPA70N to various peptides is very weak with Kd of 10-18 uM, it is very important to test whether the interactions described in the paper are functionally relevant in vivo. In particular, the authors need to carry out in vivo and/or functional analyses of the proteins with substitution/deletion in the RPA70N-binding motif in a full-length protein such as HelB or WRN.

1. The authors nicely showed different conformation of peptides bound to RPA70N. On the other hand, the authors did not mention the conformational change(s) of the RPA70N domain by itself upon the binding of the peptides. In this line, the authors nicely showed distinct conformational changes between two RPA70Ns by the binding to either BLMp1 or BLMp2 (Figure 3). It is critical to mention any structural change of a single RPA70N domain upon the binding.

2. It is also important to check the role of RPA70N-binding motifs in various proteins in vivo. The previous findings showed the role of RPA70N binding motifs in human BLM, RMI, and yeast ATRIP/Ddc1 (Deshpande et al. 2017; Shorrocks et al. 2020). These should be mentioned in the text more clearly. Much more importantly, the authors should check the binding motifs of human HelB and WRN in vivo. Different from BLM-RMI1(-TOPIII-RMI2), HelB and WRN bear only one weak RPA70N binding motif. Given the Kd of the interaction, it is very unlikely that this binding motif in the helicases is sufficient to make stable interaction with RPA in cells.

3. In the same line, it is very critical to check the binding of some full-length proteins and mutant forms of the proteins to RPA70N in vitro.

*Reviewer #2 (Recommendations for the authors):*

The structures are of high quality and provide details of the interactions. Multiple binding modes are observed based on the peptide in question. These agree with binding modes previously observed for interactors with OB-F. ITC experiments and mutagenesis are used to validate the interactions and provide quantitative information about the interactions. This study adds to the wealth of structural information available on RPA protein interactions.

The study confirms the two binding surfaces formerly identified and show that the peptides studies use these regions for interactions. The authors posit that kinking, tilting, polarity, and dimerization are features used to accommodate the peptide into the binding pockets in OB-F.

Another prominent feature highlighted by the authors is one of multivalency: where multiple RPA molecules bound on DNA could drive recruitment dimers of the interacting proteins. In this scenario, peptides can be shared between two RPA molecules and a string of RPA can be promoted through such interactions.

While these findings are interesting, this reviewer believes that these studies do not add substantially to what is already known from existing structures of OB-F protein interactions. The two binding surfaces are already established. The different modes of peptide interactions have been observed in other such structural studies.

If the authors want to build on the multivalent interactions, these results need to be substantiated with the full length proteins. For example, can the authors show that full length RPA oligomeric assemblies are driven by their target interactor? Another issue is the assumption here that the OB-F domain is freely floating around in solution. This is an incorrect assumption as OB-F interacts with other regions within the rest of the protein. Thus, the protein interactions investigated here will likely have contributions from other regions of RPA. Thus, the models proposed here that OB-F are exclusively coordinating these RPA-protein interactions are incomplete.

*Reviewer #3 (Recommendations for the authors):*

The authors have presented beautiful structures of RPA bound to peptides derived from interacting partners. The structural studies are supported by biochemical binding assays using ~22 residue peptides corresponding to wild type and carefully selected mutants to show binding. The Discussion of the paper is disappointing as it is a list of published papers and does not delineate the implications of the insight provided. So as its stands now this is a paper with beautiful structures but the implications of the data are poor. This has to be addressed. Figure 6 was challenging to sort out partially because the colour scheme is tough to see.

So how can this paper be improved? First, all the RPA70-peptide images are in different orientations so it is difficult to see where the peptides bind and to make comparisons between the different structures. In my opinion, a common orientation should be settled upon to indicate where the peptides bind. For example, as OB-folds and SH3 domains are known to be structurally similar (Agrawal, V., and Kishan, R. K. (2001) Functional evolution of two subtly different (similar) folds, BMC Struct Biol 1, 5), and SH3 domains bind PXXP-ligands in the pocket or mouth between the RT- and nSrc loops, how about an orientation that reflects this. For example:

Here I have used the OB-fold of RPA70 viewed from the side and top, and the HelB interacting residues. The position of the HelB peptide could be indicated for example in blue with orientation labeled. By using this type of approach, one can easily see where peptides bind, can they compete for binding if sites overlap, and implications of different orientations of binding can be suggested. For example, does one orientation stabilize binding to ssDNA and another enhance dissociation of RPA for example? This is important to discuss as the implication for regulation of RPA and interacting partner activity is important.

Then the authors do not make sufficient comparisons between the peptide binding pockets and again a comparison using the above figure or some other schematic should be presented.

Next, are there any mutations in either RPA70 or the interacting partners that are known to abolish function in vivo? And or be disease-related? This can further support the structural data.

How do the structural data enhance or disagree with the current biochemical data? This has been alluded to in the text but needs further elaboration.

---

## [Author Response]

Essential revisions:Please determine the interaction between both RPA and RPA-binding proteins by using full-length wild-type and mutant proteins in vitro. Given that BLMp1, BLMp2 and RMI1p could induce the oligomer formation (dimerization) of RPAs, it would be nice to focus on the binding of either BLM or RMI1 to RPA. Importantly, it is highly recommended to quantify a molar ratio of BLM/RMI1 to RPA-the authors may use gel-filtration (you may need the cross-linking to stabilize the binding) or quantitative pull-down assays or others such as low-resolution cryoEM analysis.

We thank the editor for the helpful suggestions. The BTR complex has four components BLM, TOP3a, RMI1 and RIM2 in a 2:2:2:2 ratio (Hodson et al., 2022). BLM and TOP3a are relatively large proteins with long disordered regions, we failed to obtain the intact BTR complex despite following the procedures used by Hodson et al. We’ll continue working on the preparation of full-length BTR complex. We hope that in the future we could examine the binding of RPA with full-length target proteins. Recently Spegg et al. showed that RPA has a strong propensity to assemble into dynamic condensates (phase separation), which is likely driven by RPA2 and could be stimulated by ssDNA binding (Spegg et al., 2023). More importantly, they demonstrated that RPA condensation enhances interactions with the BTR complex. The multivalent interaction we observed in the crystal structures could contribute significantly to this process in theory. Further studies with multiple approaches would be needed to delineate whether/how RPA70N functions in this dynamic process.

1. We do not ask you to do in vivo experiments (as pointed out by Reviewers #1 and #3). However, it is very important to search for literature for in vivo studies to complement your structural studies and to refer to them in the text in an organized manner.

We thank the editor for the supportive comment. To consolidate our findings, we carried out confocal imaging analysis to investigate the colocalization of RPA with WT or mutant partner proteins in HeLa cells. We also performed coimmunoprecipitation experiments to investigate the impact of mutation of interface residues on RPA-partner protein association. In general, our results agree well with previous findings; in addition, due to the availability of precise structure information, we didn’t need to use deletion mutations. When describing our results, we paid particular attention to comparing our findings with previous studies.

2. As reviewers #2 and #3 pointed out, you need to add more explanation of your experiment design and results and to discuss your findings logically. And also, you need to reformat your figures by showing RPA-peptides interaction in the same way (see comments by #3 reviewer).

We thank the editor for the very helpful comment. As suggested, we’ve added new figures to help explain the background and experiment design (new Figure 1A-1E, Figure 1—figure supplement 1A and 1B). We’ve also displayed structures determined in this study in a uniform way (new Figure 1F-1N). In addition, we summarized some of the structure information in Figure 10A for comparison.

3. Reviewer #3 provided a good example of how to show images of your 3D structure in your Figures. Please use it during your revision.

We thank the editor for the great suggestion. As suggested, we’ve added new figures to show RPA-peptide interaction in the same way (new Figure 1F-1N). For clarity, some figures displaying specific side-chain interactions were rotated to different degrees, in these figures we labeled L12 and L45 loops to mark the viewing direction.

Reviewer #1 (Recommendations for the authors):The conclusion in the paper is well supported by the results described in the paper. However, given the binding affinity of RPA70N to various peptides is very weak with Kd of 10-18 uM, it is very important to test whether the interactions described in the paper are functionally relevant in vivo. In particular, the authors need to carry out in vivo and/or functional analyses of the proteins with substitution/deletion in the RPA70N-binding motif in a full-length protein such as HelB or WRN.1. The authors nicely showed different conformation of peptides bound to RPA70N. On the other hand, the authors did not mention the conformational change(s) of the RPA70N domain by itself upon the binding of the peptides. In this line, the authors nicely showed distinct conformational changes between two RPA70Ns by the binding to either BLMp1 or BLMp2 (Figure 3). It is critical to mention any structural change of a single RPA70N domain upon the binding.

We thank the reviewer for raising this important question. Superposition of the 9 structures determined in this study with apo RPA70N showed that most of the Ca atoms of RPA70N were at nearly identical positions, with RMSD values smaller than 0.3 Å. The L12 and L45 regions displayed some conformational changes to accommodate different peptides (Figure 10B). Overall, it appears that the interaction of RPA70N with partner protein motifs relies on the movement of side chains of the conserved positively-charged and hydrophobic residues as illustrated in the main figures. We added the text to the discussion session (line #417-422).

2. It is also important to check the role of RPA70N-binding motifs in various proteins in vivo. The previous findings showed the role of RPA70N binding motifs in human BLM, RMI, and yeast ATRIP/Ddc1 (Deshpande et al. 2017; Shorrocks et al. 2020). These should be mentioned in the text more clearly. Much more importantly, the authors should check the binding motifs of human HelB and WRN in vivo. Different from BLM-RMI1(-TOPIII-RMI2), HelB and WRN bear only one weak RPA70N binding motif. Given the Kd of the interaction, it is very unlikely that this binding motif in the helicases is sufficient to make stable interaction with RPA in cells.

We thank the reviewer for the insightful suggestions. To consolidate our findings, we carried out confocal imaging analysis to investigate the colocalization of RPA with WT or mutant partner proteins in HeLa cells. We also performed coimmunoprecipitation experiments to investigate the impact of mutation of interface residues on RPA-partner protein association. Our in vivo results confirmed that the interface residues we uncovered from structural studies are critical for RPA-partner protein interaction. Although HelB only has one RPA binding motif, its affinity (4 µM) is relatively high among RPA70N binding proteins. Transient transfection of HelB-EGFP into HeLa cells resulted in bright HelB and RPA foci that colocalized in nearly 80% of cells (Figure 2H-2J,), mutation of the HelB residues involved in RPA70 binding led to much fewer foci in cells. WRN has two identical RPA binding motifs, for each motif the affinity is around 11.6 µM. Transient transfection of EGFP-WRN into HeLa cells resulted in obvious WRN and RPA foci, most of which were colocalized, especially after camptothecin (CPT) or hydroxyurea (HU) treatment (Figure 5H-5K). Transfection of a double mutant EGFP-WRN resulted in less WRN foci formation and in most cells endogenous RPA no longer formed foci, highlighting the interplay between WRN and RPA. We believe that individual RPA70 has a relatively low affinity for target proteins to prevent unwanted DNA damage response. DNA damage induced ssDNA exposure enriches RPA onto ssDNA, which would greatly enhance RPA binding to partner proteins, much like streptavidin tetramer has a higher affinity for biotin than a streptavidin monomer.

3. In the same line, it is very critical to check the binding of some full-length proteins and mutant forms of the proteins to RPA70N in vitro.

We thank the reviewer for the suggestion. Most of the RPA binding motifs are located in the disordered and flexible regions of the partner proteins. We’ve encountered difficulties in preparing intact full-length partner proteins. Instead, we performed coimmunoprecipitation analysis with full-length proteins, the results showed that mutation of interface residues reduced their binding to RPA trimer (new Figure 2K, 3N, 4H, 5L, 6K, 7E, 8F, 9H).

Reviewer #2 (Recommendations for the authors):The structures are of high quality and provide details of the interactions. Multiple binding modes are observed based on the peptide in question. These agree with binding modes previously observed for interactors with OB-F. ITC experiments and mutagenesis are used to validate the interactions and provide quantitative information about the interactions. This study adds to the wealth of structural information available on RPA protein interactions.The study confirms the two binding surfaces formerly identified and show that the peptides studies use these regions for interactions. The authors posit that kinking, tilting, polarity, and dimerization are features used to accommodate the peptide into the binding pockets in OB-F.Another prominent feature highlighted by the authors is one of multivalency: where multiple RPA molecules bound on DNA could drive recruitment dimers of the interacting proteins. In this scenario, peptides can be shared between two RPA molecules and a string of RPA can be promoted through such interactions.While these findings are interesting, this reviewer believes that these studies do not add substantially to what is already known from existing structures of OB-F protein interactions. The two binding surfaces are already established. The different modes of peptide interactions have been observed in other such structural studies.

We thank the reviewer for the insightful comments. It is true that the groove and the side pocket have been described by previous studies on p53 and DNA2. However, there are many differences as to how each protein interacts with RPA70N. We summarized some of the features in the new Figure 10A. For example, in the case of p53, the C-terminus end of the p53 peptide binds to the groove, the N-terminus end binds to the side pocket in another RPA70N. BLMp1 and RMI1 bind to RPA70N similar to p53, however their peptide direction is reversed compared to p53. In the case of DNA2, one DNA2 peptide binds both the groove and the side pocket in one RPA70N molecule; RAD9 does the same, but again the peptide direction is reversed compared to DNA2. In the case of PrimPol or yeast Ddc2, only the groove is involved in binding, ATRIP, MRE11 and ETAA1 use similar modes to bind RPA70N as PrimPol or Ddc2, nevertheless they differ a lot as to the specific side-chain arrangements since their sequences are divergent (Figure 1figure supplement 1A). In this study we’ve greatly expanded the structure information regarding RPA70N protein interactions through a systematic approach.

If the authors want to build on the multivalent interactions, these results need to be substantiated with the full length proteins. For example, can the authors show that full length RPA oligomeric assemblies are driven by their target interactor? Another issue is the assumption here that the OB-F domain is freely floating around in solution. This is an incorrect assumption as OB-F interacts with other regions within the rest of the protein. Thus, the protein interactions investigated here will likely have contributions from other regions of RPA. Thus, the models proposed here that OB-F are exclusively coordinating these RPA-protein interactions are incomplete.

We thank the reviewer for the helpful suggestions. We agree that it would be beneficial to interrogate the effect of multivalent interactions with full-length proteins. However, many of the proteins involved in this study are large protein complexes with disordered regions, it would take tremendous effort to assemble and purify these complexes in full-length form. To consolidate our findings, we expressed EGFP or FLAG-tagged partner proteins in HeLa cells to see whether they could promote RPA foci formation. HelB, BLM, WRN, ATRIP and ETTA1 formed a great number of foci and colocalized with endogenous RPA (stained with antibodies against RPA2), which is further stimulated by CPT or HU treatment (new Figure 2H-2J, 3J-3M, 5H-5K, 6F-6J, 9D-9G). Mutation of interface residues significantly weakens the foci formation and colocalization (new Figure 2H-2J, 3J-3M, 5H-5K, 6F-6J, 9D-9G). RMI1, MRE11 and RAD9 didn’t form distinct foci, but still colocalized with RPA to different degrees, which was weakened by interface mutations (new Figure 4G, 7D, 8E). Thus, it’s possible that ssDNA enriches RPA on the damage site which promotes its association with partner proteins; at the same time, RPA70N binding to partner proteins with multivalent sites in turn promotes RPA oligomeric assembly. We’ve added the text to the discussion.

We agree with the reviewer that 70N/OB-F interacts with other parts of the RPA trimer and other regions of RPA also mediate interactions with many DNA damage response proteins, which is a highly complex and dynamic process (Caldwell and Spies, 2020). For example, ETAA1 has two RPA binding motifs, one for RPA70N, and the other for RPA32C.

For this study we focused on RPA70N interacting motifs, to the best of our knowledge, when RPA binds to ssDNA, the RPA70N domain is quite dynamic (Caldwell and Spies, 2020; Deshpande et al., 2017; Kuppa et al., 2022; Pokhrel et al., 2019; Yates et al., 2018). The model we presented in old Figure 6 has been replaced with Figure 10—figure supplement 1, our intention is to explain the possible role of RPA70N in DNA damage response protein recruitment with a simplified model. We’ve now acknowledged the limitations of our study in the discussion (line #485-487).

Reviewer #3 (Recommendations for the authors):The authors have presented beautiful structures of RPA bound to peptides derived from interacting partners. The structural studies are supported by biochemical binding assays using ~22 residue peptides corresponding to wild type and carefully selected mutants to show binding. The Discussion of the paper is disappointing as it is a list of published papers and does not delineate the implications of the insight provided. So as its stands now this is a paper with beautiful structures but the implications of the data are poor. This has to be addressed. Figure 6 was challenging to sort out partially because the colour scheme is tough to see.So how can this paper be improved? First, all the RPA70-peptide images are in different orientations so it is difficult to see where the peptides bind and to make comparisons between the different structures. In my opinion, a common orientation should be settled upon to indicate where the peptides bind. For example, as OB-folds and SH3 domains are known to be structurally similar (Agrawal, V., and Kishan, R. K. (2001) Functional evolution of two subtly different (similar) folds, BMC Struct Biol 1, 5), and SH3 domains bind PXXP-ligands in the pocket or mouth between the RT- and nSrc loops, how about an orientation that reflects this. For example:Here I have used the OB-fold of RPA70 viewed from the side and top, and the HelB interacting residues. The position of the HelB peptide could be indicated for example in blue with orientation labeled. By using this type of approach, one can easily see where peptides bind, can they compete for binding if sites overlap, and implications of different orientations of binding can be suggested. For example, does one orientation stabilize binding to ssDNA and another enhance dissociation of RPA for example? This is important to discuss as the implication for regulation of RPA and interacting partner activity is important.

We sincerely appreciate the reviewer for the detailed advice to improve the presentation of the figures. As suggested, we’ve now displayed the structures determined in this study in a uniform way in new Figure 1F-1N, we’ve also labeled L12 and L45 in all figures as a reference for direction. In addition, we summarized some of the binding features in Figure 10A for comparison. RPA binds ssDNA in a strict 5’-3’ direction (Fan and Pavletich, 2012; Kolpashchikov et al., 2001) which is important for positioning the excision repair nucleases XPG, ERCC1–XPF as well as DNA2 on the DNA (de Laat et al., 1998; Zhou et al., 2015). For RPA70N the peptide binds in either direction (Figure 10A), we currently don’t have any data to show the effect of the peptide binding direction, we’ll investigate this interesting question in future studies.

Then the authors do not make sufficient comparisons between the peptide binding pockets and again a comparison using the above figure or some other schematic should be presented.

We thank the reviewer for pointing this out. We’ve added more figures to show the binding pockets (new Figure 1B, 1C, 1E) and we’ve also compared RPA70N from different structures to show its conformational changes (Figure 10B).

Next, are there any mutations in either RPA70 or the interacting partners that are known to abolish function in vivo? And or be disease-related? This can further support the structural data.

We thank the reviewer for the excellent suggestion. Many of the DNA damage response proteins are related to cancer or rare genetic diseases like Bloom syndrome, Werner syndrome. We’ve checked the COSMIC and cBioPortal databases, R31 in RPA70N is mutated to H or C in some cancer patients, which might be related to its role in protein protein interaction. With ITC, we showed the R31H mutation reduced the interaction between RPA70N and interacting peptides (Figure 2—figure supplement 1G, Figure 3figure supplement 2F, Figure 4—figure supplement 1E, Figure 5—figure supplement 1E, Figure 8—figure supplement 1C, Figure 9—figure supplement 1C). Currently, we don’t know how this mutation affects cells, which warrants more investigation in future studies. We didn’t find meaningful point mutations specifically related to the RPA70N binding motifs in BLM or WRN from Bloom syndrome or Werner syndrome patients. To support the structural data, we carried out confocal imaging analysis to investigate the colocalization of RPA with WT or mutant partner proteins in HeLa cells. We’ve also performed coimmunoprecipitation experiments to investigate the impact of mutation of interface residues on RPA-partner protein association. We now show that the interface residues we’ve identified in the structures are crucial for the recruitment of DNA damage response proteins to the RPA foci and many of the partner proteins are able to promote RPA foci formation, which is closely linked to DNA damage response.

How do the structural data enhance or disagree with the current biochemical data? This has been alluded to in the text but needs further elaboration.

We thank the reviewer for raising this important question. In general, we believe the structural data enhances the understanding and application of the current biochemical data. First, our systematic structural characterization of RPA70N-target protein interaction provides a molecular basis for partner protein recruitment by RPA70N and rich information for the DNA damage response field. For example, with structure-guided mutations it is possible to precisely disrupt the interaction between RPA70N and target proteins without causing unwanted side effects. Second, through systematic comparison, we show that RPA70N utilizes its two binding interfaces in various modes. The multivalent interaction we’ve observed in the crystal structures could contribute significantly to the condensation-driven DNA damage response processes as means for rapid signal amplification. Third, our structure information helps to understand how a small domain like RPA70N recognizes diverse target sequences with low sequence homology, which is still challenging for algorithms to predict accurately. Fourth, RPA proteins are often overexpressed in cancer cells due to increased replication stress, great efforts have been made to develop inhibitors to reduce RPA-target protein association to curtail cancer growth and increase the efficacy of chemotherapy drugs (Glanzer et al., 2013; Glanzer et al., 2014). The structural data presented in this study would definitely benefit this process.

References:

Bochkareva, E., Kaustov, L., Ayed, A., Yi, G.S., Lu, Y., Pineda-Lucena, A., Liao, J.C., Okorokov, A.L., Milner, J., Arrowsmith, C.H., and Bochkarev, A. (2005). Single-stranded DNA mimicry in the p53 transactivation domain interaction with replication protein A. Proc Natl Acad Sci U S A *102*, 1541215417. 10.1073/pnas.0504614102.

Caldwell, C.C., and Spies, M. (2020). Dynamic elements of replication protein A at the crossroads of DNA replication, recombination, and repair. Crit Rev Biochem Mol Biol *55*, 482-507. 10.1080/10409238.2020.1813070.

de Laat, W.L., Appeldoorn, E., Sugasawa, K., Weterings, E., Jaspers, N.G., and Hoeijmakers, J.H. (1998). DNA-binding polarity of human replication protein A positions nucleases in nucleotide excision repair. Genes Dev *12*, 2598-2609. 10.1101/gad.12.16.2598.

Deshpande, I., Seeber, A., Shimada, K., Keusch, J.J., Gut, H., and Gasser, S.M. (2017). Structural Basis of Mec1-Ddc2-RPA Assembly and Activation on Single-Stranded DNA at Sites of Damage. Mol Cell *68*, 431-445 e435. 10.1016/j.molcel.2017.09.019.

Fan, J., and Pavletich, N.P. (2012). Structure and conformational change of a replication protein A heterotrimer bound to ssDNA. Genes Dev *26*, 2337-2347. 10.1101/gad.194787.112.

Feldkamp, M.D., Frank, A.O., Kennedy, J.P., Patrone, J.D., Vangamudi, B., Waterson, A.G., Fesik, S.W., and Chazin, W.J. (2013). Surface reengineering of RPA70N enables cocrystallization with an inhibitor of the replication protein A interaction motif of ATR interacting protein. Biochemistry *52*, 65156524. 10.1021/bi400542z.

Glanzer, J.G., Carnes, K.A., Soto, P., Liu, S., Parkhurst, L.J., and Oakley, G.G. (2013). A small molecule directly inhibits the p53 transactivation domain from binding to replication protein A. Nucleic Acids Res *41*, 2047-2059. 10.1093/nar/gks1291.

Glanzer, J.G., Liu, S., Wang, L., Mosel, A., Peng, A., and Oakley, G.G. (2014). RPA inhibition increases replication stress and suppresses tumor growth. Cancer Res *74*, 5165-5172. 10.1158/00085472.CAN-14-0306.

Guilliam, T.A., Brissett, N.C., Ehlinger, A., Keen, B.A., Kolesar, P., Taylor, E.M., Bailey, L.J., Lindsay, H.D., Chazin, W.J., and Doherty, A.J. (2017). Molecular basis for PrimPol recruitment to replication forks by RPA. Nat Commun *8*, 15222. 10.1038/ncomms15222.

Hodson, C., Low, J.K.K., van Twest, S., Jones, S.E., Swuec, P., Murphy, V., Tsukada, K., Fawkes, M., Bythell-Douglas, R., Davies, A., et al. (2022). Mechanism of Bloom syndrome complex assembly required for double Holliday junction dissolution and genome stability. Proc Natl Acad Sci U S A *119*. 10.1073/pnas.2109093119.

Kolpashchikov, D.M., Khodyreva, S.N., Khlimankov, D.Y., Wold, M.S., Favre, A., and Lavrik, O.I. (2001). Polarity of human replication protein A binding to DNA. Nucleic Acids Res *29*, 373-379.10.1093/nar/29.2.373.

Kuppa, S., Deveryshetty, J., Chadda, R., Mattice, J.R., Pokhrel, N., Kaushik, V., Patterson, A., Dhingra, N., Pangeni, S., Sadauskas, M.K., et al. (2022). Rtt105 regulates RPA function by configurationally stapling the flexible domains. Nat Commun *13*, 5152. 10.1038/s41467-022-32860-6.

Pokhrel, N., Caldwell, C.C., Corless, E.I., Tillison, E.A., Tibbs, J., Jocic, N., Tabei, S.M.A., Wold, M.S., Spies, M., and Antony, E. (2019). Dynamics and selective remodeling of the DNA-binding domains of RPA. Nat Struct Mol Biol *26*, 129-136. 10.1038/s41594-018-0181-y.

Souza-Fagundes, E.M., Frank, A.O., Feldkamp, M.D., Dorset, D.C., Chazin, W.J., Rossanese, O.W., Olejniczak, E.T., and Fesik, S.W. (2012). A high-throughput fluorescence polarization anisotropy assay for the 70N domain of replication protein A. Anal Biochem *421*, 742-749.10.1016/j.ab.2011.11.025.

Spegg, V., Panagopoulos, A., Stout, M., Krishnan, A., Reginato, G., Imhof, R., Roschitzki, B., Cejka, P., and Altmeyer, M. (2023). Phase separation properties of RPA combine high-affinity ssDNA binding with dynamic condensate functions at telomeres. Nat Struct Mol Biol *30*, 451-462. 10.1038/s41594023-00932-w.

Yates, L.A., Aramayo, R.J., Pokhrel, N., Caldwell, C.C., Kaplan, J.A., Perera, R.L., Spies, M., Antony, E., and Zhang, X. (2018). A structural and dynamic model for the assembly of Replication Protein A on single-stranded DNA. Nat Commun *9*, 5447. 10.1038/s41467-018-07883-7.

Zhou, C., Pourmal, S., and Pavletich, N.P. (2015). Dna2 nuclease-helicase structure, mechanism and regulation by Rpa. *eLife 4*. 10.7554/*eLife*.09832.